# Great ape cognition is structured by stable cognitive abilities and predicted by developmental conditions

Manuel Bohn [1] ✉, Johanna Eckert[1], Daniel Hanus[1], Benedikt Lugauer[2], Jana Holtmann[2] & Daniel B. M. Haun [1,3]

Great ape cognition is used as a reference point to specify the evolutionary origins of complex cognitive abilities, including in humans. This research often assumes that great ape cognition consists of cognitive abilities (traits) that account for stable differences between individuals, which change and develop in response to experience. Here, we test the validity of these assumptions by assessing repeatability of cognitive performance among captive great apes (*Gorilla gorilla*, *Pongo abelii*, *Pan paniscus*, *Pan troglodytes*) in five tasks covering a range of cognitive domains. We examine whether individual characteristics (age, group, test experience) or transient situational factors (life events, testing arrangements or sociality) influence cognitive performance. Our results show that task-level performance is generally stable over time; four of the five tasks were reliable measurement tools. Performance in the tasks was best explained by stable differences in cognitive abilities (traits) between individuals. Cognitive abilities were further correlated, suggesting shared cognitive processes. Finally, when predicting cognitive performance, we found stable individual characteristics to be more important than variables capturing transient experience. Taken together, this study shows that great ape cognition is structured by stable cognitive abilities that respond to different developmental conditions.

In their quest to understand the evolution of cognition, anthropologists, psychologists and cognitive scientists face a major obstacle: cognition does not fossilize. Instead of studying the cognitive abilities of, for example, extinct early hominins directly, we have to rely on inferences. We can, for example, study fossilized skulls and crania to approximate brain size and structure and use this information to infer cognitive abilities[1,2]. We can also study the material culture left behind by extinct species and try to infer its cognitive complexity[3–5]. Yet, the archaeological record is sparse and only goes back so far. Thus, additionally, we rely on backwards inference about a last common ancestor on the basis of the phylogenetically informed comparison of extant species. The so-called comparative method is one of the most fruitful approaches to investigating cognitive evolution. If species A and B both show cognitive ability X, the last common ancestor of A and B most probably also had ability X[6–9]. In this way, similarities and differences between species are used to make inferences about points of divergence in the evolutionary tree as well as about external drivers of this divergence. Following this approach, comparing humans to non-human great apes has been highly productive and provides the empirical basis for numerous theories about human cognitive evolution[10–15].

[1]Department of Comparative Cultural Psychology, Max Planck Institute for Evolutionary Anthropology, Leipzig, Germany. [2]Wilhelm Wundt Institute of Psychology, Leipzig University, Leipzig, Germany. [3]Leipzig Research Centre for Early Child Development, Leipzig University, Leipzig, Germany. ✉e-mail: manuel_bohn@eva.mpg.de

The use of cross-species comparisons to make backwards inferences about human cognitive evolution relies on a particular view of the nature and structure of great ape cognition. Cognition is seen as structured in the form of cognitive abilities that account for stable differences between individuals and which evolve and develop in response to enduring social and environmental conditions. Such differences in cognitive abilities are involved in generating variation in the behaviour on which selection can act[16,17]. Without a stable cognitive basis that is systematically linked to behaviour, cognitive evolution is not possible: at least, not in the way it is commonly theorized about[18]. In this study, we seek to provide empirical answers to a series of questions asking whether this view on great ape cognition holds. Alternatively, performance in cognitive tasks could be largely determined by transient situational factors and not capture stable abilities of individuals. Because cognitive abilities cannot directly be observed, asking questions about the structure of great ape cognition inevitably comes with asking questions about the tools—experimental tasks—that are used to measure it.

The first question is whether studies on great ape cognition produce robust results: inferences about the cognitive abilities of great apes—as a clade, species, or group—should remain the same across repeated studies with different individuals or follow predictable patterns in studies with the same individuals. This is a critical requirement to build theories around the results of cross-species comparisons. In practice, the robustness of aggregated results is implicitly assumed but rarely tested[19–22].

The second question is whether there are stable differences between individuals and whether tasks commonly used in great ape cognition research are able to reliably measure them. This is a prerequisite to investigate the extent to which differences between individuals in one ability covary with differences in other abilities to map out the internal structure of great ape cognition[18,23–25]. Once again, in practice, this is simply assumed to be the case but rarely tested empirically.

Finally, we ask which social and environmental conditions influence cognition. That is, we look for individual characteristics or everyday experiences that predict performance in our measures of cognitive ability. On the one hand, such predictive relationships inform us about the nature of cognitive performance: is it heavily influenced by transient and situational factors or malleable to long-term experiences? On the other hand, they inform us about the contexts in which cognitive abilities emerge and are the cornerstone for theorizing about the ontogeny and phylogeny of cognitive abilities[26,27]. To summarize, so far, we know too little about the structure of great ape cognition to judge the validity of the comparative method as a way to study the origins of of human cognition.

There are several studies that provide a more comprehensive picture of one or more aspects of the nature and structure of great ape cognition[24,28–32]. Herrmann and colleagues[33] tested more than 100 great apes (chimpanzees and orangutans) and human children in various tasks covering numerical, spatial and social cognition. The results indicated pronounced group-level differences between great apes and humans in the social but not the spatial or numerical domain. Furthermore, relationships between the tasks pointed to a different internal structure of cognition, with a distinct social cognition factor for humans but not great apes[34,35]. Völter and colleagues[36] focused on the structure of executive functions. Using a multi-trait multi-method approach[37], they developed a new test battery to assess memory updating, inhibition and attention shifting in chimpanzees and human children. Overall, they found low correlations between tasks and, thus, no clear support for structures put forwards by theoretical models built around adult human data.

Beyond great apes, there have been numerous attempts to investigate the structure of cognition in other animals[24]. In many cases, test batteries have been used to find evidence for a 'general cognitive ability', that is, a correlation of individual performance across tasks[38–42]. Such studies found consistent individual differences across

two or more tasks in various species and taxa (for example, insects[43,44], rodents[45–47] and birds[48,49]). Some even correlated these differences with individual characteristics such as sex or relatedness[43,44,47].

Despite their contributions to understanding the nature and structure of animal and great ape cognition, these studies suffer from one or more of the shortcomings outlined above: it is unclear whether the results are robust. If the same individuals were tested again, would the results licence the same conclusions about absolute differences between species? Furthermore, the psychometric properties of the tasks are unknown and it is thus unclear if, for example, low correlations between tasks reflect a genuine lack of shared cognitive processes or simply measurement imprecision. Most importantly, which characteristics and experiences predict cognitive performance remains unclear. Establishing such a link is essential if we want to understand cognitive abilities and the driving forces behind their emergence and development.

The studies reported here address the shortcomings outlined above and seek to solidify the empirical grounds on which the use of the comparative method for investigating the evolution of human cognition rests. For 1.5 years, every 2 weeks, we administered a set of five cognitive tasks (Fig. 1) to the same population of great apes ($n = 43$). The tasks spanned cognitive domains and were based on published procedures widely used in comparative psychology. As a test of social cognition, we included a gaze-following task[50]. To assess causal reasoning abilities, we had a direct causal inference and an inference by exclusion task[51]. Numerical cognition was tested using a quantity discrimination task[52]. Finally, as a test of executive functions, we included a delay of gratification task[53] (second half of the study only, below). In the first half, we used a different measure of executive functions[54]. This task, however, failed to produce meaningful results (see the Supplementary Material and Supplementary Figs. 8 and 9 for details).

In addition to the cognitive data, we continuously collected 14 variables that capture stable and variable aspects of our participants and their lives and used this to predict inter- and intra-individual variation in cognitive performance. These predictors included (1) stable differences between individuals (group, age, sex, rearing history, experience with research), (2) differences that varied within and between individuals (rank, sickness, sociality), (3) differences that varied with group membership (time spent outdoors, disturbances, life events) and (4) differences in testing arrangements (presence of observers, participation in unrelated studies on the same day and since the last time point).

Data collection was split into two phases; after phase 1 (14 data collection time points), we analysed the data and registered the results (https://osf.io/7qyd8). Phase 2 lasted for another 14 time points and served to replicate and extend phase 1. This approach allowed us to test (1) how robust task-level results are, (2) how reliable individual differences are measured and how stable they are over time, (3) how individual differences are structured and (4) what predicts cognitive performance.

## Results

### Robustness of task-level performance
As a first step, we asked whether the average performance of a given sample at a time is robust, that is, whether we could assume to find a similar average performance for a given sample of individuals if we repeated the identical tasks. We assessed robustness in two ways: first, whenever there was a level of performance expected by chance (that is, 50% correct), we checked whether the 95% confidence interval for the mean proportion correct overlapped with chance. Second, we assessed temporal robustness using structural equation modelling (SEM), in particular, latent state models (see Methods and Supplementary Fig. 6 for details). These models partition the observed performance variable at a given time point into a latent state variable (time-specific true score variable) and a measurement-error variable. The mean of the latent

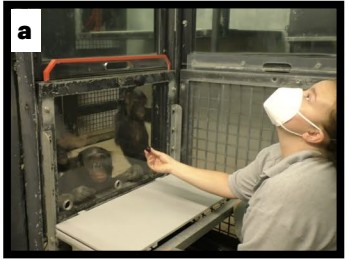
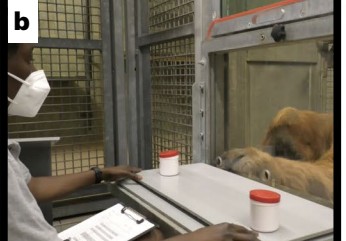
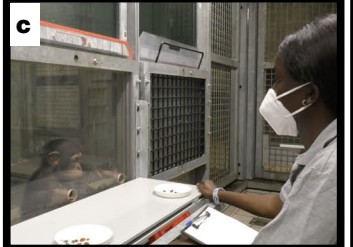
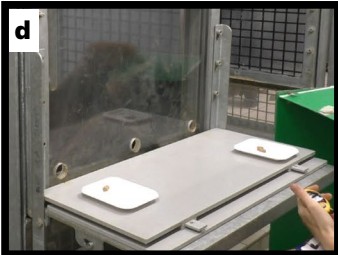

**e**

| Session 1 | | | Session 2 | | | |
|---|---|---|---|---|---|---|
| Gaze following (2 trials) | Direct causal inference and inference by exclusion (12 trials each, intermixed) | Gaze following (2 trials) | Gaze following (2 trials) | Quantity discrimination (12 trials) | Delay of gratification (12 trials) | Gaze following (2 trials) |

**Fig. 1 | Setup used for the five tasks. a**, For gaze following, the experimenter looked to the ceiling. We coded if the ape followed gaze. **b**, For direct causal inference, food was hidden in one of two cups, the baited cup was shaken (food produced a sound) and apes had to choose the shaken cup to get food. For inference by exclusion, food was hidden in one of two cups. The empty cup was shaken (no sound), so apes had to choose the non-shaken cup to get food. **c**, For quantity discrimination, small pieces of food were presented on two plates (five versus seven items); we coded if subjects chose the larger amount. **d**, For delay of gratification (only phase 2), to receive a larger reward, the subject had to wait and forgo a smaller, immediately accessible reward. **e**, Order of task presentation, trial numbers and organization of tasks into sessions. In both phases, we ran the two sessions on two separate days.

state variable for the first time point of each phase was fixed at zero and we assessed average change across time by asking whether the 95% credible intervals (CrI) for the latent state means of subsequent time points overlapped with zero (that is, the mean of the first time point).

Task-level performance was largely robust or followed clear temporal patterns. Figure 2 visualizes the proportion of correct responses for each task; Fig. 3a shows the latent state means for each task and phase. The direct causal inference and quantity discrimination tasks were the most robust: in both cases, performance was different from chance across both phases with no apparent change over time. The rate of gaze following declined at the beginning of phase 1 but then settled on a low but stable level until the end of phase 2. This pattern was expected given that following the experimenter's gaze was never rewarded: neither explicitly with food nor by bringing something interesting to the participant's attention. The inference by exclusion task showed an inverse pattern with task-level performance being at chance level for most of phase 1, followed by a small but steady increase throughout phase 2 so that from time point 6 in phase 2 onwards, performance was consistently different from the first time point of that phase. These temporal patterns most likely reflect training (or habituation) effects that are a consequence of repeated testing. Performance in the delay of gratification task (phase 2 only) was more variable but within the same general range for the whole testing period. In sum, despite these exceptions, performance was very robust in that time points generally supported the same task-level conclusions. For example, Fig. 2 shows that performance in the direct causal inference task was clearly above chance at all time points and, on a descriptive level, consistently higher compared to the inference by exclusion task. Thus, the tasks appeared well suited to study task-level performance.

### Reliability of individual-level measurements
The reliability of a measure is defined as the proportion of true score variance to its observed total variance. That is, a reliable measure captures interindividual differences with precision (that is, perfect reliability corresponds to measurement without measurement error) and is expected to produce similar results if repeated under identical conditions. In practice, however, there may be a trade-off between aggregate and individual-level measurement goals: an observation that has been coined the 'reliability paradox'[55].

As a first step towards investigating individual differences, we inspected retest correlations of our five tasks. For that, we correlated the performance at the different time points in each task (Fig. 4). Correlations were generally high, some even exceptionally high, for animal

cognition standards[22]. As expected, values were higher for more proximate time points[56]. The quantity discrimination task had lower correlations compared to the other tasks.

However, on the basis of retest correlations alone, we cannot say whether lower correlations reflect higher measurement error (low reliability) or interindividual differences in (true) change of performance across time (low stability). To tease these components apart, we turned again to the latent state models mentioned above. For each time point, we estimated a latent state variable (time-specific true score variable) using two test halves as indicators. These test halves were constructed by splitting the trials of each task per time point into two parallel subgroups. Thereby, the models allow us to estimate the reliability of the respective test halves (Methods and section patent state models in the Supplementary Material for details). We interpreted reliability estimates in the following way: acceptable, 0.7; good, 0.8 and high 0.9. Please note that these estimates are for test halves; the reliability of the full test would be higher.

Figure 3b shows that reliability was generally good (roughly 0.75) for all tasks at all time points, except for the quantity discrimination task that had reliability estimates fluctuating around 0.5. Thus, the lower retest correlations for quantity discrimination most probably reflect low reliability instead of individual changes in cognitive performance across time. We will return to this point again in the next section. Taken together, these results indicate that most tasks reliably measured differences between individuals.

As a final note, the results show that task-level robustness does not imply individual-level stability, and vice versa. The quantity discrimination task showed robust task-level performance above chance (Fig. 2) but relatively poor reliability (Fig. 3b). In other words, even though task-level performance was similar at all time points, differences between individuals were measured with low precision. In contrast, task-level performance in the inference by exclusion and gaze-following tasks changed over time, with satisfactory measurement precision and moderate to high stability of true interindividual differences (next section).

### Structure and stability of interindividual differences
Next, we investigated the structure of individual differences. In contrast to earlier work[34], with 'structure' we do not exclusively mean the relationship between different cognitive tasks. As mentioned in the introduction, we start with a more basic question: do individual differences in a given task reflect differences in cognitive ability (for example, ability to make causal inferences) that persist over time or rather

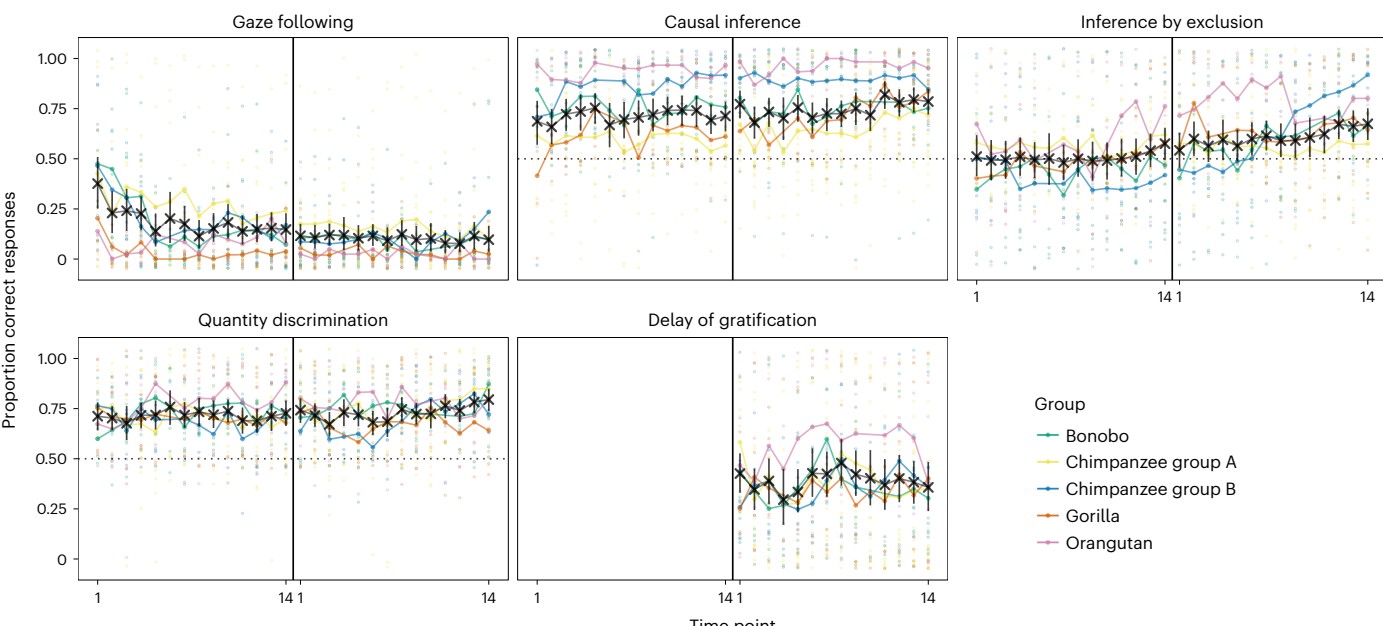

**Fig. 2 | Results from the five cognitive tasks across time points.** Black crosses show mean performance at each time point across all individuals in the sample at that time point (with 95% confidence interval). The sample size varied between time points and can be found in Supplementary Fig. 1. Coloured dots show mean performance by group. Light dots show individual means per time point. Dashed lines show chance level whenever applicable. The vertical black line marks the transition between phases 1 and 2.

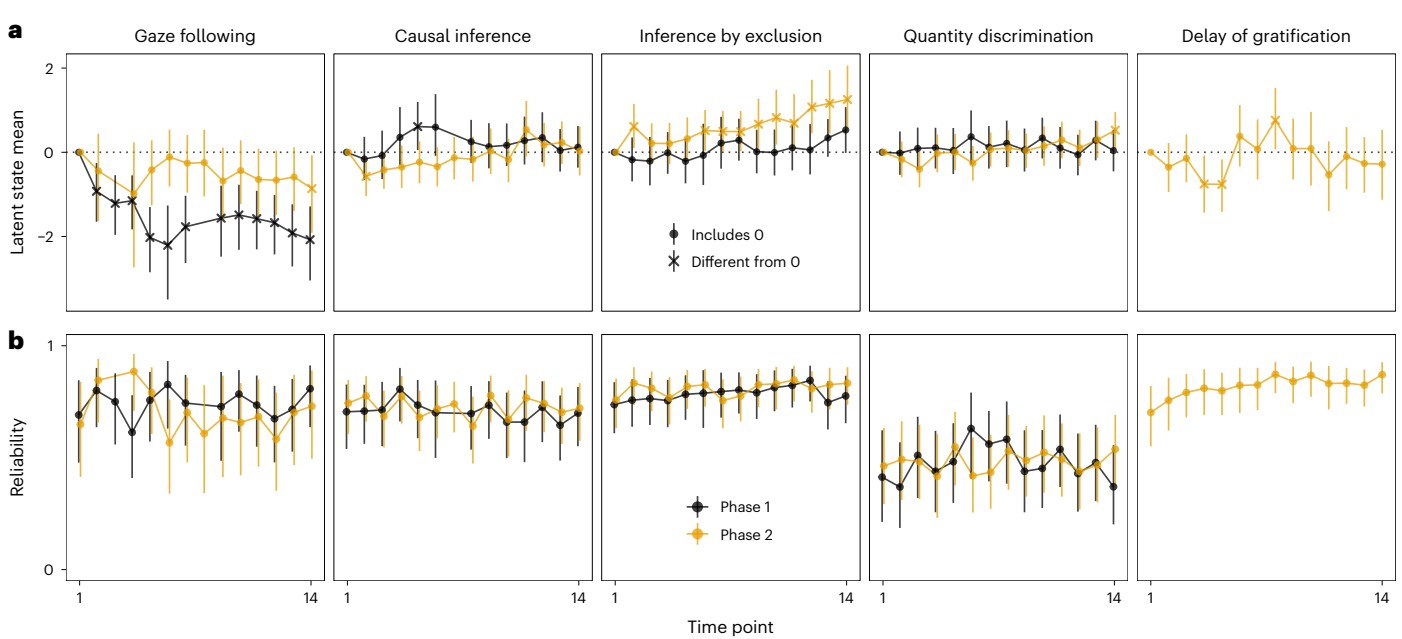

**Fig. 3 | Stability and reliability estimates based on latent state models.** **a**, Latent state means for each time point by task and phase estimated via latent state models. The first time point is set to zero as a reference point. Shape denotes whether the 95% CrI included zero (dashed line). The sample size varied between time points and can be found in Supplementary Fig. 1. **b**, Corresponding reliability estimates. Points show mean of the posterior distribution with 95% CrI.

differences in transient factors (for example, motivation or attentiveness) that vary from time point to time point. The former would indicate that individuals (true scores) are ranked similarly across time points, while the latter would predict fluctuations in ranks.

To quantify to what extent stable or variable differences between individuals explain performance, we used latent state-trait (LST) models that partitioned the observed performance score into a latent trait variable, a latent state residual variable and measurement error[57–59]. We assume stable latent traits, such that one can think of a latent trait as a stable cognitive ability (for example, the ability to make causal inferences) and latent state residuals as variables capturing the effect of occasion-specific psychological conditions (for example, being more or less attentive or motivated). The sum of the latent trait and the latent state residual variable corresponds to the true score of cognitive performance at a specific time point (latent state variable). We report additional models that account for the temporal structure of the data in the Supplementary Material (Supplementary Note).

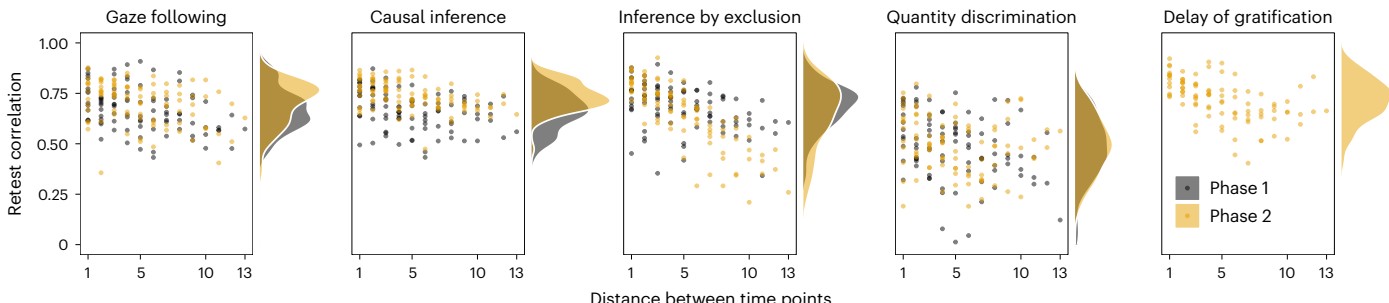

**Fig. 4 | Retest correlation coefficients are plotted against the temporal distance between the testing time points.** The colour shows the phase. The side shows the distribution of retest Pearson correlation coefficients.

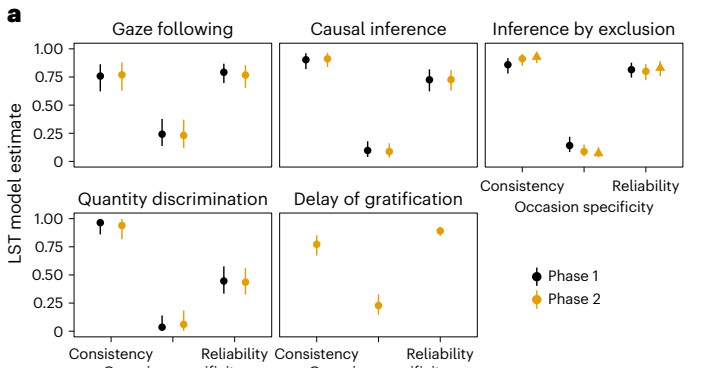

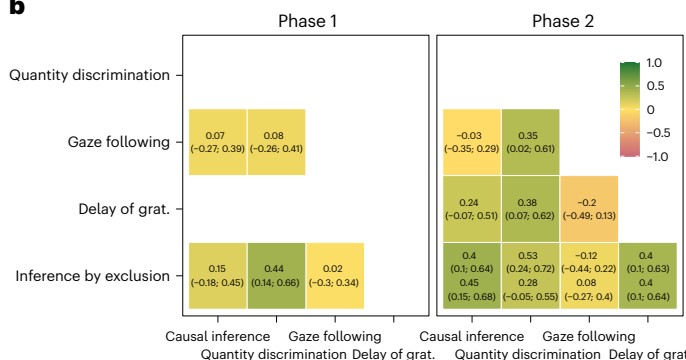

**Fig. 5 | Latent state-trait model estimates and correlations between latent traits across tasks. a**, Mean estimates from latent state-trait models for phases 1 and 2 with 95% CrI based on data from n = 43 participants. Consistency refers to the proportion of (measurement-error-free) variance in performance explained by stable trait differences. Occasion specificity refers to the proportion of true variance explained by variable state residuals. Reliability refers to the proportion of true score variance to variance in raw scores. For inference by exclusion,

different shapes show estimates for different parts of phase 2 (see main text for details). **b**, Correlations between latent traits based on pairwise latent state-trait models between tasks with 95% CrI. Bold correlations have CrI not overlapping with zero. Inference by exclusion has one value per part in phase 2. The models for quantity discrimination and direct causal inference showed a poor fit and are not reported here (see Supplementary Material for details).

True individual differences were largely stable across time. Across tasks, more than 75% of the reliable variance (true interindividual differences) was accounted for by latent trait differences and less than 25% by occasion-specific variation between individuals (Fig. 5a). The good reliability estimates (more than 0.75 for most tasks; Fig. 5a) show that these latent variables accounted for most of the variance in raw test scores, with the quantity discrimination task being an exception (reliability 0.47). Reflecting back on the results reported above, we can now say that the, relatively speaking, lower correlations between time points in the quantity discrimination task indicate a higher degree of measurement error rather than variable individual differences. In fact, once measurement error is accounted for, consistency estimates for the quantity discrimination task were close to one, reflecting highly stable true differences between individuals.

Next, we compared the estimates for the two phases of data collection. We found estimates for consistency (proportion of true score variance due to latent trait variance) and occasion specificity (proportion of true score variance due to state residual variance) to be similar for the two phases. For inference by exclusion, the LST model did not fit the data from phase 2 well (see Supplementary Material for details). Therefore, we divided phase 2 into two parts (time points 1–8 and 9–14) and estimated a separate trait for each part. All estimates were similar for both parts (Fig. 5a), and the two traits were highly correlated (r = 0.82). Together with the latent state model results reported in the Robustness of task-level performance section, this indicates that the increase in group-level performance in phase 2 was probably driven

by a relatively sudden improvement of a few individuals, mostly from the chimpanzee B group (Fig. 2). These individuals quickly improved in performance halfway through phase 2 and retained this level for the rest of the study.

Finally, we investigated the relationship between latent traits. We asked whether individuals with high abilities in one domain also have higher abilities in another. We fit pairwise LST models that modelled the correlation between latent traits for two tasks (two models for inference by exclusion in phase 2). In phase 1, the only substantial correlation (that is, coefficients indicated medium to large effects[60] and their 95% CrI did not include zero) was between quantity discrimination and inference by exclusion. In phase 2, this finding was replicated, and, in addition, four more correlations turned out to be substantial (Fig. 5b). One reason for this increase was the inclusion of the delay of gratification task. Across phases, correlations involving the gaze-following task were the closest to zero, with quantity discrimination in phase 2 being an exception. Taken together, the overall pattern of results indicates substantial shared variance between tasks, except for gaze following.

## Predictability of individual differences

The results thus far indicate that individual differences originate from stable differences between individuals in cognitive abilities that persist across time points. Differences in ability outweigh fluctuations due to transient, occasion-specific factors such as attentiveness or motivation. An alternative pattern would arise when time point-specific variation in for example, attentiveness or motivation would be responsible for

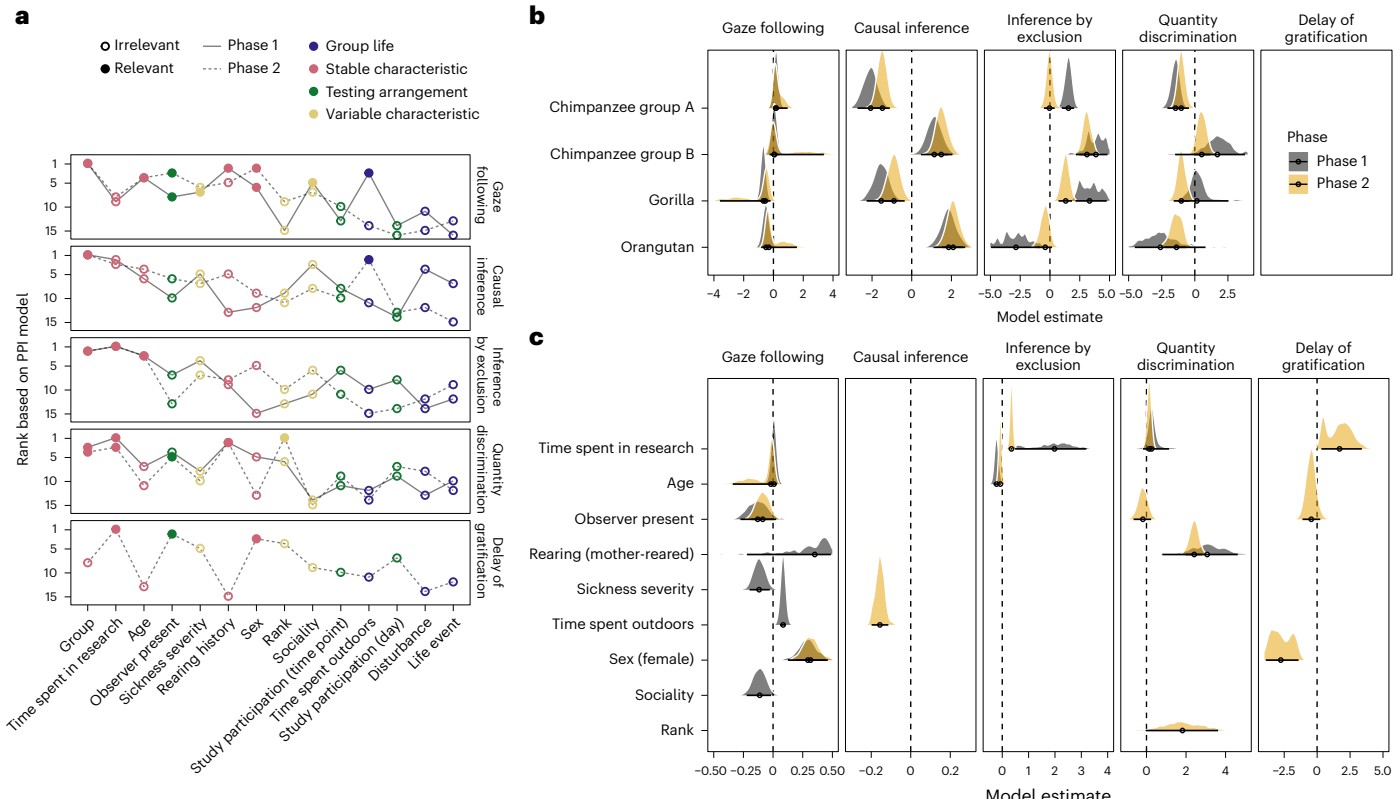

**Fig. 6 | Predictors of cognitive performance across tasks and time points.**
**a**, Ranking of predictors based on the projection predictive inference model for the five tasks in the two phases. The order (left to right) is based on average rank across phases. Solid points indicate predictors selected as relevant. The colour of the points shows the category of the predictor. Line type denotes the phase. **b**, Posterior model estimates for group for tasks for which it was selected as relevant. **c**, Posterior model estimates for the remaining selected predictors for each task based on data from $n = 43$ participants. Points show means with 95% CrI. Colour denotes phase. For categorical predictors, the estimate gives the difference compared to the reference level (Bonobo for group, no observer for observer, hand-reared for rearing, male for sex).

differences in performance between individuals. Of course, there can be stable differences between individuals in attentiveness and motivation, in which case they would influence performance in a consistent way over time and presumably also across tasks[61–63]. The distinction we want to make here is between transient and stable factors influencing cognitive performance.

In the last set of analyses, we sought to explain the origins of individual differences. That is, we analysed whether inter- and intra-individual variation in cognitive performance in the tasks could be predicted by non-cognitive variables that captured (1) stable differences between individuals (group, age, sex, rearing history, experience with research), (2) differences that varied within and between individuals (rank, sickness, sociality), (3) differences that varied with group membership (time spent outdoors, disturbances, life events) and (4) differences in testing arrangements (presence of observers, study participation on the same day and since the last time point). We collected these predictor variables using a combination of directed observations and caretaker questionnaires.

This large set of potentially relevant predictors poses a variable selection problem. Thus, in our analysis, we sought to find the smallest number of predictors (main effects only) that allowed us to accurately predict performance in the cognitive tasks. We chose the projection predictive inference approach because it provides an excellent trade-off between model complexity and accuracy[64–66]. The outcome of this analysis is a ranking of the different predictors in terms of how important they are to predicting performance in a given task. Furthermore, for each predictor, we get a qualitative assessment of whether it makes a substantial contribution to predicting performance in the task or not.

Predictors capturing stable individual characteristics were ranked highest and selected as relevant most often (Fig. 6a). The three highest-ranked predictors belonged to this category. This result fits well with the LST model results reported above, in which we saw that most of the variance in performance could be traced back to stable trait differences between individuals. Here we saw that performance was best predicted by variables that reflect stable characteristics of individuals. This suggests that stable characteristics partially cause selective development that leads to differences in cognitive abilities. The tasks with the highest occasion-specific variance (gaze following and delay of gratification, Fig. 5a) were also those for which the most time point-specific predictors were selected. The quantity discrimination task did not fit this pattern in phase 2; even though the LST model indicated that only a very small portion of the variance in performance was occasion-specific, four time point-specific variables were selected to be relevant.

The most important predictor was group. Differences between groups were not systematic in that one group consistently outperformed the others across tasks. Furthermore, group differences could not be collapsed into species differences as the two chimpanzee groups varied largely independently of one another (Fig. 6b). Predictors that were selected more than once influenced performance in variable ways. The presence of observers always had a negative effect on performance. The more time an individual had been involved in research during their lifetime, the better performance was. On the other hand, while the rate of gaze following increased with age in phase 1, performance in the inference by exclusion task decreased. Females were more likely to follow gaze than males, but males were more likely to wait for the larger

reward in the delay of gratification task. Finally, time spent outdoors had a positive effect on gaze following but a negative effect on direct causal inference (Fig. 6b).

In sum, individual characteristics were most predictive of cognitive performance. In most cases, the corresponding predictors were selected as relevant in both phases. The influence of time point-specific predictors was less consistent: except for the presence of an observer in the gaze-following task, none of the variable predictors was selected as relevant in both phases. To avoid misinterpretation, this suggests that cognitive performance was influenced by temporal variation in group life, testing arrangements and variable characteristics; however, the way this influence exerts itself was either less consistent or less pronounced (or both) compared to the influence of stable characteristics.

It is important to note, however, that in terms of absolute variance explained, the largest portion was accounted for by a random intercept term in the model (not shown in Fig. 5) that simply captured the identity of the individual (Supplementary Figs. 21–29). This suggests that idiosyncratic developmental processes and/or genetic predispositions, which operate on a much longer time scale than what we captured in the present study, were responsible for most of the variation in cognitive performance.

## Discussion

This study aimed to test the assumptions of robustness, stability, reliability and predictability that underlie much of comparative research and theorizing about cognitive evolution. We repeatedly tested a large sample of great apes in five tasks covering a range of different cognitive domains. We found task-level performance to be robust for most tasks so that conclusions drawn on the basis of one testing occasion mirrored those on other occasions. Most of the tasks measured differences between individuals in a reliable and stable way, making them suitable to study individual differences. Using SEMs, we found that individual differences in performance were largely explained by traits: that is, stable differences in cognitive abilities between individuals. Furthermore, we found systematic relationships between cognitive abilities. When predicting variation in cognitive performance, we found stable individual characteristics (for example, group or time spent in research) to be the most important. Variable predictors were also found to be influential at times but less systematically.

At first glance, the results send a reassuring message: most of the tasks we used produced robust task-level results and captured individual differences in a reliable and stable way. However, this did not apply to all tasks. As noted above, in the Supplementary Material, we report on a rule-switching task[54] that produced neither stable nor reliable results (Supplementary Figs. 8 and 9). The quantity discrimination task was robust on a task level but did not measure individual differences reliably. We draw two conclusions on the basis of this pattern. First, replicating studies, even if it is with the same animals, should be an integral part of primate cognition research[67–69]. Second, for individual differences research, it is crucial to assess the psychometric properties (for example, reliability) of the measures involved[70]. If this step is omitted, it is difficult to interpret studies, especially when they produce null results. It is important to note that the sample size in the current study was large compared to other comparative studies (median sample size across studies of seven)[67]. With smaller sample sizes, task-level estimates are probably more variable and thus more likely to produce false-positive or -negative conclusions[71,72]. Small samples in comparative research usually reflect the resource limitations of individual labs. Pooling resources in large-scale collaborative projects such as ManyPrimates[73,74] will thus be vital to corroborate findings.

Continuing on this theme, the data reported here would be exciting to explore for species differences. For example, the descriptive results shown in Fig. 2 indicate that orangutans performed best in the non-social tasks but worse in the social task. However, we are hesitant to interpret such findings because of the small sample sizes per species and the substantial differences in sample size between species. Consequently, it is impossible to distinguish individual-level from species-level variation.

Given their good psychometric properties, our tasks offer insights into the structure of great ape cognition. We used SEM to partition reliable variance in performance into stable (trait) and variable (state residual) differences between individuals. We found traits to explain more than 75% of the reliable variance across tasks. This suggests that the patterns in performance we observed mainly originate from stable differences in cognitive abilities. This finding does not mean there cannot be developmental change over longer time periods. In fact, for the inference by exclusion task, we saw a relatively abrupt change in performance for some individuals, which stabilized on an elevated level, suggesting a sustained change in cognitive ability.

We found systematic relationships between traits estimated via LST models for the different tasks. Correlations tended to be higher among the non-social tasks compared to when the gaze-following task was involved, which could be taken to indicate shared cognitive processes. However, we feel such a conclusion would be premature and require additional evidence from more tasks and larger sample sizes[34]. One possibility is that stable, domain-general psychological processes, such as attentiveness or motivation, are responsible for the shared variance. Cognitive modelling could be used to explicate the processes involved in each task. Shared processes could be probed by comparing models that make different assumptions[75,76].

The finding that stable differences in cognitive abilities explained most of the variation between individuals was also corroborated by the analyses focused on the predictability of performance. We found that predictors that captured stable individual characteristics (for example, group, time spent in research, age, rearing history) were more likely to be selected as relevant predictors. Aspects of everyday experience or testing arrangements that would influence performance on particular time points and thus increase the proportion of occasion-specific variation (for example, life events, disturbances, participating in other tests) were ranked as less important. Despite this general pattern, there was variation across tasks in which individual characteristics were selected to be relevant. For example, rearing history was an important predictor for quantity discrimination and gaze following but less so for the other three tasks (Fig. 6a). Group, the overall most important predictor, exerted its influence differently across tasks. Orangutans, for example, outperformed the other groups in direct causal inference but were the least likely to follow gaze. Together with the finding that the random intercept term explained the largest proportion of variance in performance across tasks, this pattern suggests that the cognitive abilities underlying performance in the different tasks respond to different, although sometimes overlapping, external conditions that together shape the individual's developmental environment.

Our results also address a very general matter. Comparative psychologists often worry, or are told they should worry, that their results can be explained by mechanistically simpler associative learning processes[77]. Often, such explanations are theoretically plausible and rarely disproved empirically[78]. The present study speaks to this issue in so far as we created the conditions for such associative learning processes to potentially unfold. Great apes were tested by the same experimenter in the same tasks, using differential reinforcement and the same counterbalancing for hundreds of trials. However, a steady increase in performance, uniform over individuals, did not show. This does not take away the theoretical possibility that associative learning accounts for improved performance over time on isolated tasks. In fact, we are agnostic as to whether or not a particular learning account might explain our results (or parts of them) and invite others to further analyse the data provided here.

## Conclusion

The present study put the implicit assumptions underlying much of comparative research on cognitive evolution involving great apes to an empirical test. While we found reassuring results in terms of task-level stability and reliability of the measurement of individual differences, we also pointed out the importance of explicitly questioning and testing these assumptions, ideally in large-scale collaborative projects. Our results paint a picture of great ape cognition in which variation between individuals is predicted and explained by stable individual characteristics that respond to different, although sometimes overlapping, developmental conditions. Hence, an ontogenetic perspective is not auxiliary but fundamental to studying cognitive diversity across species. We hope these results contribute to a more solid and comprehensive understanding of the nature and origins of great ape and human cognition as well as provide useful methodological guidance for future comparative research.

## Methods

### Participants

A total of 43 great apes participated at least once in one of the tasks. This included eight bonobos (*Pan paniscus*, three females, age 7.30 to 39), 24 chimpanzees (*Pan troglodytes*, 18 females, age 2.60 to 55.90), six gorillas (*Gorilla gorilla*, four females, age 2.70 to 22.60) and five orangutans (*Pongo abelii*, four females, age 17 to 41.20). The overall sample size at the different time points ranged from 22 to 43 for the different species.

Apes were housed at the Wolfgang Köhler Primate Research Centre located in Zoo Leipzig, Germany. They lived in groups, with one group per species and two chimpanzee groups (A and B). Studies were non-invasive and adhered to the legal requirements in Germany. Animal husbandry and research complied with the European Association of Zoos and Aquaria Minimum Standards for the Accommodation and Care of Animals in Zoos and Aquaria as well as the World Association of Zoos and Aquariums Ethical Guidelines for the Conduct of Research on Animals by Zoos and Aquariums. Participation was voluntary, all food was given in addition to the daily diet and water was available ad libitum throughout the study. The study was approved by an internal ethics committee at the Max Planck Institute for Evolutionary Anthropology.

### Procedure

Apes were tested in familiar sleeping or test rooms by a single experimenter. Whenever possible, they were tested individually. The basic setup comprised a sliding table positioned in front of a clear Plexiglas panel with three holes in it. The experimenter sat on a small stool and used an occluder to cover the sliding table (Fig. 1).

The tasks we selected are based on published procedures and are commonly used in the field of comparative psychology. Example videos for each task can be found in the associated online repository (https://github.com/ccp-eva/laac/tree/master/videos).

**Gaze following.** The gaze-following task was modelled after a study by Bräuer and colleagues[50]. The experimenter sat opposite the ape and handed over food at a constant pace. That is, the experimenter picked up a piece of food, briefly held it out in front of her face and then handed it over to the participant. After a predetermined (but varying) number of food items had been handed over, the experimenter again picked up a food item, held it in front of her face and then looked up (that is, moving her head up: Fig. 1a). The experimenter looked to the ceiling; no object of particular interest was placed there. After 10 s, the experimenter looked down again, handed over the food and the trial ended. We coded whether the participant looked up during the 10 s interval. Apes received eight gaze-following trials. We assume that participants look up because they assume that the experimenter's attention is focused on a potentially noteworthy object.

**Direct causal inference.** The direct causal inference task was modelled after a study by Call[51]. Two identical cups, each with a lid, were placed left and right on the table (Fig. 1b). The experimenter covered the table with the occluder, retrieved a piece of food, showed it to the ape and hid it in one of the cups outside the participant's view. Next, the experimenter removed the occluder, picked up the baited cup and shook it three times, which produced a rattling sound. Next, the cup was put back in place, the sliding table pushed forwards and the participant made a choice by pointing to one of the cups. If they picked the baited cup, their choice was coded as correct and they received the reward. If they chose the empty cup, they did not. Participants received 12 trials. The location of the food was counterbalanced; six times in the right cup and six times in the left. Direct causal inference trials were intermixed with inference by exclusion trials (below). We assume that apes locate the food by reasoning that the food, a solid object, causes the rattling sound and, therefore, must be in the shaken cup.

**Inference by exclusion.** Inference by exclusion trials were also modelled after the study by Call[51] and followed a very similar procedure compared to direct causal inference trials. After covering the two cups with the occluder, the experimenter placed the food in one of the cups and covered both with the lid. Next, they removed the occluder, picked up the empty cup and shook it three times. In contrast to the direct causal inference trials, this did not produce any sound. The experimenter then pushed the sliding table forwards and the participant made a choice by pointing to one of the cups. Correct choice was coded when the baited (non-shaken) cup was chosen. If correct, the food was given to the ape. There were 12 inference by exclusion trials intermixed with direct causal inference trials. The order was counterbalanced: six times the left cup was baited, six times the right. We assume that apes reason that the absence of a sound suggests that the shaken cup is empty. Because they saw a piece of food being hidden, they exclude the empty cup and infer that the food is more likely to be in the non-shaken cup.

**Quantity discrimination.** For this task, we followed the general procedure of Hanus and colleagues[52]. Two small plates were presented left and right on the table (Fig. 1c). The experimenter covered the plates with the occluder and placed five small food pieces on one plate and seven on the other. Then they pushed the sliding table forwards, and the participant made a choice. We coded as correct when the subject chose the plate with the larger quantity. Participants always received the food from the plate they chose. There were 12 trials, six with the larger quantity on the right and six on the left (order counterbalanced). We assume that apes identify the larger of the two food amounts on the basis of discrete quantity estimation.

**Delay of gratification.** This task replaced the switching task in phase 2. The procedure was adapted from Rosati and colleagues[53]. Two small plates, including one and two pieces of pellet, were presented left and right on the table. The experimenter moved the plate with the smaller reward forward, allowing the subject to choose immediately, while the plate with the larger reward was moved forwards after a delay of 20 s. We coded whether the subject selected the larger delayed reward (correct choice) or the smaller immediate reward (incorrect choice) as well as the waiting time in cases where the immediate reward was chosen. Subjects received 12 trials, with the side on which the immediate reward was presented counterbalanced. We assume that, to choose the larger reward, apes inhibit choosing the immediate smaller reward.

**Interrater reliability.** A second coder unfamiliar to the purpose of the study coded 15% of all time points (four out of 28) for all tasks. Reliability was good to excellent. Gaze following showed 92% agreement ($\kappa = 0.64$) and direct causal inference 99% agreement ($\kappa = 0.98$); inference by exclusion showed 99% agreement ($\kappa = 0.99$); quantity discrimination

showed 99% agreement ($\kappa = 0.97$) and delay of gratification showed 98% agreement ($\kappa = 0.97$).

## Data collection

We collected data in two phases. Phase 1 started on 1 August 2020, lasted until 5 March 2021, and included 14 time points. Phase 2 started on 26 May 2021, and lasted until 4 December 2021, and also had 14 time points. Phase 1 also included a strategy switching task. However, because it did not produce meaningful results, we replaced it with the delay of gratification task. Details and results can be found in the Supplementary Material available (Supplementary Figs. 8 and 9).

One time point meant running all tasks with all participants. Within each time point, the tasks were organized in two sessions (Fig. 1e). Session 1 started with two gaze-following trials. Next was a pseudorandomized mix of direct causal inference and inference by exclusion trials with 12 trials per task but no more than two trials of the same task in a row. At the end of session 1, there were again two gaze-following trials. Session 2 also started with two gaze-following trials, followed by quantity discrimination and strategy switching (phase 1) or delay of gratification (phase 2). Finally, there were two further gaze-following trials. The order of tasks was the same for all subjects. So was the positioning of food items within each task. The two sessions were usually spread out across two adjacent days. The interval between two time points was planned to be 2 weeks. However, it was not always possible to follow this schedule, so some intervals were longer or shorter. Supplementary Fig. 1 shows the timing and spacing of the time points.

In addition to the data from the cognitive tasks, we collected data for a range of predictor variables. Predictors could vary with the individual (stable individual characteristics were group, age, sex, rearing history and time spent in research), vary with individual and time point (variable individual characteristics were rank, sickness and sociality), vary with group membership (group life, for example, time spent outdoors, disturbances and life events) or vary with the testing arrangements and thus with individual, time point and session (testing arrangements, such as presence of observers, study participation on the same day and since the last time point). Most predictors were collected by means of a diary that the animal caretakers filled out on a daily basis. Here, the caretakers were asked a range of questions about the presence of a predictor and its severity. Other predictors were based on direct observations. A detailed description of the predictors and how they were collected can be found in the Supplementary Material.

## Analysis

In the following, we provide an overview of the analytical procedures we used. We encourage the reader to consult the Supplementary Material available online for additional details. We had two overarching questions. On the one hand, we were interested in the cognitive measures and the relationships between them. That is, we asked how robust performance was on a task level, how stable individual differences were and how reliable the measures were. We also investigated relationships between the different tasks. We used SEM[79,80] to address these questions.

Our second question was, which predictors explained variability in cognitive performance? Here, we wanted to see which of the predictors we recorded were most important to predict performance over time. This is a variable selection problem (selecting a subset of variables from a larger pool) and we used projection predictive inference for this[66].

**SEM.** We used SEM[79,80] to address the reliability and stability of each task, as well as relationships between tasks. SEMs allowed us to partition the variance in performance into latent variable (true score) variance and measurement-error variance. Latent variables are estimated using multiple observed indicators (here, two test halves: below). Longitudinal data for each task was modelled with a latent state and a LST model[57–59]. All of the models were estimated as normal-ogive

graded response models[81,82] due to the ordinal nature of the indicators. For each task and time point we split the trials in two test halves, which served as indicators for a common latent construct. Owing to only few different observed values and skewed distributions of the sum score for each test half, indicators were modelled as ordered categorical variables using a probit link function. That is, the models assume a continuous latent ability underlying the discrete responses, with an increasing probability of more correctly solved trials with increasing ability.

Formally speaking, the observed categorical variables $Y_{it}$ for test half $i$ at time point $t$ result from a categorization of unobserved continuous latent variables that underlie the observed categorical variables. In the latent state models, $Y_{it}^*$ is decomposed into into a latent state variable $S_t$ and a measurement-error variable $\epsilon_{it}$ (ref. 83). At each time point $t$, the two latent variables $Y_{1t}^*$ and $Y_{2t}^*$ are assumed to capture a common latent state variable $S_t$. To test for possible mean changes of ability across time, the means of the latent state variables were freely estimated (assuming invariance of the threshold parameters $\kappa_{sit}$ across time).

As an estimate of reliability (Rel), we computed the proportion of true score variance (Var) relative to the total variance of the continuous latent variables $Y_{it}^*$:

$$\text{Rel}\left(Y_{it}^*\right) = \frac{\text{Var}\left(S_t\right)}{\text{Var}\left(S_t\right) + \text{Var}\left(\epsilon_{it}\right)} = \frac{\text{Var}\left(S_t\right)}{\text{Var}\left(S_t\right) + 1}$$

For the LST model, the continuous latent variable $Y_{it}^*$ was decomposed into a latent trait variable $T_{it}$, a latent state residual variable $\zeta_{it}$ and a measurement-error variable. The latent trait variables $T_{it}$ are time-specific dispositions, that is, they capture the expected value of the latent state (that is, true score) variable for an individual at time $t$ across all possible situations the individual might experience at time $t$ (refs. 58,84). The state residual variables $\zeta_{it}$ capture the deviation of a momentary state from the time-specific disposition $T_{it}$. We assumed that latent traits were stable across time. In addition, we assumed common latent trait and state residual variables across the two test halves, which leads to the following measurement equation for parcel $i$ at time point $t$:

$$Y_{it}^* = T + \zeta_t + \epsilon_{it}$$

Here, $T$ is a stable (time-invariant) latent trait variable, capturing stable interindividual differences. The state residual variable $\zeta_t$ captures time-specific deviations of the respective true score from the trait variable at time $t$, and thereby captures deviations from the trait due to situation or person–situation interaction effects. $\epsilon_{it}$ denotes a measurement-error variable, with $\epsilon_{it} \approx N(0,1) \; \forall \; i,t$. This allowed us to compute the following variance components.

Consistency (Con) refers to the proportion of true variance (that is, measurement-error-free variance) that is due to true interindividual stable trait differences.

$$\text{Con}\left(Y_{it}^*\right) = \frac{\text{Var}\left(T\right)}{\text{Var}\left(T\right) + \text{Var}\left(\zeta_t\right)}$$

Occasion specificity (OS) refers to the proportion of true variance (that is, measurement-error-free variance) that is due to true interindividual differences in the state residual variables (that is, occasion-specific variation not explained by the trait).

$$\text{OS}\left(Y_{it}^*\right) = 1 - \text{Con}\left(Y_{it}^*\right) = \frac{\text{Var}\left(\zeta_t\right)}{\text{Var}\left(T\right) + \text{Var}\left(\zeta_t\right)}$$

As state residual variances $\text{Var}(\zeta_t)$ were set equal across time, $\text{OS}(Y_{it}^*)$ is constant across time (as well as across item parcels $i$).

To investigate associations between cognitive performance in different tasks, the LST models were extended to multi-trait models. Owing to the small sample size, we could not combine all tasks in a single, structured model. Instead, we assessed relationships between tasks in pairs.

We used Bayesian estimation techniques to estimate the models. In the Supplementary Material, we report the prior settings used for estimation as well as the restrictions we imposed on the model parameters. We justify these settings by means of simulation studies described in the Supplementary Note.

**Projection predictive inference.** The selection of relevant predictor variables constitutes a variable selection problem, for which a range of different methods are available, for example, shrinkage priors[85]. We chose to use projection predictive inference because it provides an excellent trade-off between model complexity and accuracy[64,66], especially when the goal is to identify a minimal subset of predictors that yield a good predictive model[65].

The projection predictive inference approach can be viewed as a two-step process. The first step consists of building the best predictive model possible, called the reference model. In the context of this work, the reference model is a Bayesian multilevel regression model with repeated measurements nested in apes, fit using the package brms[86], including all 14 predictors and a random intercept term for the individual (R notation, `DV ≅ predictors + (1 | subject)`). Note that this reference model only included main effects and no interactions between predictors. Including interactions would have increased the number of predictors to consider exponentially.

In the second step, the goal is to replace the posterior distribution of the reference model with a simpler distribution. This is achieved via a forwards step-wise addition of predictors that decrease the Kullback–Leibler divergence from the reference model to the projected model.

The result of the projection is a list containing the best model for each number of predictors from which the final model is selected by inspecting the mean log-predictive density (elpd) and root-mean-squared error (r.m.s.e.). The projected model with the smallest number of predictors is chosen, which shows similar predictive performance as the reference model.

We built separate reference models for each phase and task and ran them through the above-described projection predictive inference approach. The dependent variable for each task was the cognitive performance of the apes, that is, the number of correctly solved trials per time point and task. The model for the delay of gratification task was only estimated once (phase 2).

We used the R package projpred[87], which implements the aforementioned projection predictive inference technique. The predictor relevance ranking is measured by the leave-one-out cross-validated elpd and r.m.s.e. To find the optimal submodel size, we inspected summaries and the plotted trajectories of the calculated elpd and r.m.s.e.

The order of relevance for the predictors and the random intercept (together called terms) is created by performing a forwards search. The term that decreases the Kullback–Leibler divergence between the reference model's predictions and the projection's predictions the most goes into the ranking first. The forwards search is then repeated *n* times to get a more robust selection. We chose the final model by inspecting the predictive utility of each projection. To be precise, we chose the model with *p* terms where *p* depicts the number of terms at the cut off between the term that increases the elpd and the term that does not increase the elpd by any substantial amount. To get a useful predictor ranking, we manually delayed the random intercept (and random slope for time point for gaze following) term to the last position in the predictor selection process. The random intercept delay is needed because if the random intercept were not delayed, it would soak up almost all of the variance of the dependent variable before the predictors are allowed to explain some amount of the variance themselves.

## Reporting summary
Further information on research design is available in the Nature Portfolio Reporting Summary linked to this article.

## Data availability
All data can be found in the following public repository: https://github.com/ccp-eva/laac. The same repository also contains example videos for the different tasks.

## Code availability
All analysis code needed to reproduce the results and figures reported in the paper and the Supplementary Material can be found in the following public repository: https://github.com/ccp-eva/laac.

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

## Acknowledgements

We thank D. Olaoba, A. Wolff and N. Eisenbrenner for the data collection. We are very grateful to M. Allritz for his helpful comments on an earlier version of the paper. Furthermore, we thank all keepers at the Wolfgang Köhler Primate Research Centre for their help conducting this study. We received no specific funding for this work.

## Author contributions

M.B. did the conceptualization and formal analysis, prepared the original draft and reviewed and edited the paper. J.E. did the conceptualization, prepared the original draft and reviewed and edited the paper. D.H. did the conceptualization, prepared the original draft and reviewed and edited the paper. B.L. did the formal analysis, prepared the original draft and reviewed and edited the paper. J.H. did the formal analysis, prepared the original draft and reviewed and edited the paper. D.B.M.H. did the conceptualization and reviewed and edited the paper.

## Funding

## Competing interests

The authors declare no competing interests.

## Additional information

**Correspondence and requests for materials** should be addressed to Manuel Bohn.

# Reporting Summary

## Statistics

For all statistical analyses, confirm that the following items are present in the figure legend, table legend, main text, or Methods section.

| n/a | Confirmed | |
|---|---|---|
| ☐ | ☒ | The exact sample size (*n*) for each experimental group/condition, given as a discrete number and unit of measurement |
| ☐ | ☒ | A statement on whether measurements were taken from distinct samples or whether the same sample was measured repeatedly |
| ☒ | ☐ | The statistical test(s) used AND whether they are one- or two-sided *Only common tests should be described solely by name; describe more complex techniques in the Methods section.* |
| ☐ | ☒ | A description of all covariates tested |
| ☐ | ☒ | A description of any assumptions or corrections, such as tests of normality and adjustment for multiple comparisons |
| ☐ | ☒ | A full description of the statistical parameters including central tendency (e.g. means) or other basic estimates (e.g. regression coefficient) AND variation (e.g. standard deviation) or associated estimates of uncertainty (e.g. confidence intervals) |
| ☒ | ☐ | For null hypothesis testing, the test statistic (e.g. $F$, $t$, $r$) with confidence intervals, effect sizes, degrees of freedom and $P$ value noted *Give P values as exact values whenever suitable.* |
| ☐ | ☒ | For Bayesian analysis, information on the choice of priors and Markov chain Monte Carlo settings |
| ☐ | ☒ | For hierarchical and complex designs, identification of the appropriate level for tests and full reporting of outcomes |
| ☐ | ☒ | Estimates of effect sizes (e.g. Cohen's *d*, Pearson's *r*), indicating how they were calculated |

*Our web collection on statistics for biologists contains articles on many of the points above.*

## Software and code

Policy information about availability of computer code

| | |
|---|---|
| Data collection | No software was used to collect the data |
| Data analysis | R version 4.2; R packages used for inferential statistics: brms version 2.17; projpred version 2.1.2; Structural Equation Models were implemented in MPlus version 8.4. |

For manuscripts utilizing custom algorithms or software that are central to the research but not yet described in published literature, software must be made available to editors and reviewers. We strongly encourage code deposition in a community repository (e.g. GitHub). See the Nature Portfolio guidelines for submitting code & software for further information.

## Data

Policy information about availability of data

All manuscripts must include a data availability statement. This statement should provide the following information, where applicable:
- Accession codes, unique identifiers, or web links for publicly available datasets
- A description of any restrictions on data availability
- For clinical datasets or third party data, please ensure that the statement adheres to our policy

All data and analysis scripts can be found in the associated online repository (https://github.com/ccp-eva/laac).

## Human research participants

Policy information about studies involving human research participants and Sex and Gender in Research.

| | |
|---|---|
| Reporting on sex and gender | *Use the terms sex (biological attribute) and gender (shaped by social and cultural circumstances) carefully in order to avoid confusing both terms. Indicate if findings apply to only one sex or gender; describe whether sex and gender were considered in study design whether sex and/or gender was determined based on self-reporting or assigned and methods used. Provide in the source data disaggregated sex and gender data where this information has been collected, and consent has been obtained for sharing of individual-level data; provide overall numbers in this Reporting Summary. Please state if this information has not been collected. Report sex- and gender-based analyses where performed, justify reasons for lack of sex- and gender-based analysis.* |
| Population characteristics | *Describe the covariate-relevant population characteristics of the human research participants (e.g. age, genotypic information, past and current diagnosis and treatment categories). If you filled out the behavioural & social sciences study design questions and have nothing to add here, write "See above."* |
| Recruitment | *Describe how participants were recruited. Outline any potential self-selection bias or other biases that may be present and how these are likely to impact results.* |
| Ethics oversight | *Identify the organization(s) that approved the study protocol.* |

Note that full information on the approval of the study protocol must also be provided in the manuscript.

# Field-specific reporting

Please select the one below that is the best fit for your research. If you are not sure, read the appropriate sections before making your selection.

☐ Life sciences   ☒ Behavioural & social sciences   ☐ Ecological, evolutionary & environmental sciences

For a reference copy of the document with all sections, see nature.com/documents/nr-reporting-summary-flat.pdf

# Behavioural & social sciences study design

All studies must disclose on these points even when the disclosure is negative.

| | |
|---|---|
| Study description | Quantitative data obtained from behavioral experiments |
| Research sample | A total of 43 great apes participated at least once in one of the tasks. This included 8 Bonobos (3 females, age 7.30 to 39), 24 Chimpanzees (18 females, age 2.60 to 55.90), 6 Gorillas (4 females, age 2.70 to 22.60), and 5 Orangutans (4 females, age 17 to 41.20). Apes were housed at the Wolfgang Köhler Primate Research Center located in Zoo Leipzig, Germany. |
| Sampling strategy | We included all great apes living at Leipzig Zoo who were willing to participate in research. |
| Data collection | Data was collected via non-invasive behavioral experiments using materials such as cups and plates. A human experimenter presented the materials and recorded the responses on a coding sheet. In addition, data collection was video recorded and a second coder coded a subset of trials for inter-rater reliability. The experimenter was not blind to the conditions.<br>Participation was voluntary, participants were not deprived of food or water and all food received during the experiments was given in addition to the regular daily diet. |
| Timing | Phase one lasted from August 1st, 2020 until March 5th, 2021; phase 2 lasted from May 26th, 2021 to December 4th, 2021 |
| Data exclusions | No data was excluded |
| Non-participation | This was a longitudinal study and not all participants from the above mentioned pool participated at all time points. The sample size at each time point ranged from 22 to 43 individuals. |
| Randomization | This was an individual differences study, thus, all participants received the tasks in the same order. |

# Reporting for specific materials, systems and methods

We require information from authors about some types of materials, experimental systems and methods used in many studies. Here, indicate whether each material, system or method listed is relevant to your study. If you are not sure if a list item applies to your research, read the appropriate section before selecting a response.

## Materials & experimental systems

| n/a | Involved in the study |
|-----|----------------------|
| ☒ | Antibodies |
| ☒ | Eukaryotic cell lines |
| ☒ | Palaeontology and archaeology |
| ☐ | ☒ Animals and other organisms |
| ☒ | Clinical data |
| ☒ | Dual use research of concern |

## Methods

| n/a | Involved in the study |
|-----|----------------------|
| ☒ | ChIP-seq |
| ☒ | Flow cytometry |
| ☒ | MRI-based neuroimaging |

# Animals and other research organisms

Policy information about studies involving animals; ARRIVE guidelines recommended for reporting animal research, and Sex and Gender in Research

Laboratory animals
bonobos (Pan paniscus), chimpanzees (Pan troglodytes), gorillas (Gorilla gorilla), orangutans (Pongo abelii).

Wild animals
The study did not involve wild animals

Reporting on sex
Sex was considered as a predictor variable used to explain individual differences and was determined based on external genitalia.

Field-collected samples
The study did not involve samples collected from the field

Ethics oversight
The study was approved by an internal ethics committee at the Max Planck Institute for Evolutionary Anthropology.

Note that full information on the approval of the study protocol must also be provided in the manuscript.

