## [Peer Review File · Nature Ecology & Evolution]

Peer Review Information

Journal: Nature Ecology & Evolution

Manuscript Title: Great ape cognition is structured by stable cognitive abilities and predicted by developmental conditions

Corresponding author name(s): Manuel Bohn

Editorial Notes:

Reviewer Comments & Decisions:

Decision Letter, initial version:

26th August 2022

Dear Dr Bohn,

Thank you very much for your enquiry about submitting your manuscript "Probing the structure, stability, and predictability of great ape cognition" to Nature Ecology & Evolution. Because you provided the full manuscript rather than just a summary, we were able to evaluate it in full, and I am happy to say that we have decided that we can send it out for peer review.

At this stage, we ask that you resubmit it as a full Article, along with all of the author information (and suggested/excluded reviewers if desired), using this submission link:

[REDACTED]

We also ask that you upload two forms as related manuscript files. The forms are described below:

We are trying to improve the quality of methods and statistics reporting in our papers. To that end, we are asking all life sciences authors to complete two items: an editorial policy checklist that verifies compliance with all required editorial policies and a reporting summary that collects information on experimental design and reagents.

Reporting summary: <https://www.nature.com/documents/nr-reporting-summary.pdf>

Editorial policy checklist: <https://www.nature.com/documents/nr-editorial-policy-checklist.pdf>

Please note that these forms must be downloaded and completed in Adobe Reader rather than opened in a web browser. If you would like to reference the guidance text as you complete the template, please access these flattened versions at www.nature.com/authors/policies/availability.html

Please complete the entire Reporting Summary checklist. If any of the points from the checklist are not addressed in the manuscript, you can revise the paper after this round of peer review.

2Please note that if you have selected double-blind peer review, you should not include author names on the form.

If you have any questions, please feel free to contact me.

[REDACTED]

Decision Letter, first revision:

14th November 2022

Dear Dr Bohn,

Your manuscript entitled "Probing the structure, stability, and predictability of great ape cognition" has now been seen by four reviewers, whose comments are copied below. Thank you again for your patience while we recruited an additional reviewer with statistical expertise. The reviewers have raised a number of concerns which we would like to see addressed in a revised manuscript before we can reach a final decision regarding publication in Nature Ecology & Evolution.

We therefore invite you to revise your manuscript taking into account all reviewer comments. Please highlight all changes in the manuscript text file.

* If you have not done so already please begin to revise your manuscript so that it conforms to our Article format instructions at <http://www.nature.com/natecolevol/info/final-submission>. Refer also to any guidelines provided in this letter.

* Include a revised version of any required reporting checklist. It will be available to referees (and, potentially, statisticians) to aid in their evaluation if the manuscript goes back for peer review. A

2revised checklist is essential for re-review of the paper.

[REDACTED]

Nature Ecology & Evolution is committed to improving transparency in authorship. As part of our efforts in this direction, we are now requesting that all authors identified as 'corresponding author' on published papers create and link their Open Researcher and Contributor Identifier (ORCID) with their account on the Manuscript Tracking System (MTS), prior to acceptance. ORCID helps the scientific community achieve unambiguous attribution of all scholarly contributions. You can create and link your ORCID from the home page of the MTS by clicking on 'Modify my Springer Nature account'. For more information please visit www.springernature.com/orcid.

[REDACTED]

Reviewer expertise:

Reviewer #1: Repeatability in animal cognition tasks

Reviewer #2: Wild ape cognition

Reviewer #3: General comparative animal cognition

Reviewer #4: SEM and latent state-trait models

Reviewers' comments:

3Reviewer #1 (Remarks to the Author):

Review of MS NATECOLEVOL-220817321A: Probing the structure, stability, and predictability of great ape cognition

Before I say anything else, I would like to apologize to the authors for sending this in a bit late. I was settling into my new office (I am on sabbatical).

This paper describes a study of 43 great apes. The authors tested whether several cognitive traits, each measured on multiple occasions, in these apes was stable over time at the level of the group, whether individual differences in these traits were reliable, and whether they were stable within the individual/unaffected by internal factors. What I appreciated about this study was that the authors did a nice job of choosing data analytic approaches that were suitable for answering the questions that they sought to address. Finally, the writing, and explanations of their methods, was clear for the most part. All that said there were some aspects of this manuscript that I thought could use work.

First, to my mind, this paper seemed more method-oriented than problem-oriented (see, e.g., Platt, 1964). In other words, I think the authors would be much better off focusing on what questions about cognition in these species they wish to address and what different findings would mean in this regard. I am not sure how much revising this will take, but it will require some thought.

Continuing, although I cannot speak for the authors, I was under the impression that what they had in mind is that studies of cognition in primates rested on several assumptions, but that these assumptions were not tested, and that the goal of their study was to test these assumptions. If so, that is fine provided others had not tested these assumptions before or there were still gaps in this literature. Related to this, one key paper that the authors did not cite was that by Kaufman et al. (2019). The authors may also wish to extend their literature review to go beyond great apes. I don't think every last paper needs to be cited, but more could be if they wish to make the case for this study.

Second, I failed to grasp the importance of the analyses/question about the "stability of group-level performance". How is it critical whether one species or group or another shows the same age trajectories (or sex differences, etc.) than others, or even a clear pattern, for this kind of trait? Many traits change over time, much in the same way as physical characteristics, although some do not, but that does not make one more or less trait-like. If there is some expectation, grounded in trait theory and cognitive psychology, of why one should see the same pattern of change across groups or species, then the authors did not clearly spell out why one should have these expectations.

Third, the most important finding was at the end of the Result section (275-280). The best evidence that these are traits or trait-like is that most of the variance was not due to transient, situational factors, but consistent between-individual variance. This needs more prominence.

Fourth, I disagree that their study addresses the question of whether the results of their study and that of others can be understood using associative learning (see paragraph starting on line 350). The second statement (lines 352-353) is simply an assertion with nothing backing it. Just because the same experiment was present, differential reinforcement was used, etc., does not mean that a steady increase, let alone a uniform increase across individuals, should occur. Here, again, the paper by Platt is an outstanding guide to working out what hypotheses/explanations your study can and cannot rule out. I would omit this text.

In addition to these major points, I noticed some minor matters that the authors should address. I will refer to each by line numbers.

4l 31-32: The sentence here is awkward and should be rewritten.
l 33: The use of the word "licensed" here is odd.
l 33-34: This is not clear.
l 35: It is probably best to refer to "relationships" in this sentence.
l 53: I would omit the phrase "in time"
l 53: The comparative method could be brought up sooner. A brief mention and citation of the other approaches that you refer to with relevant references is all that is necessary.
l 55: The word "backward" is needed here.
l 58: Do not hyphenate "nonhuman". Also, change "such as" to "this".
l 72-73: I would rephrase this to avoid the word "prototypical".
l 76: References are needed for the end of this sentence.
l 78: I think it would be better to refer to "assumptions" and stick to that. Something other than "problems" would be advisable in any event, and please be consistent with the terminology throughout, i.e., edits to lines 92 and 93, and so on.
l 100: Should "below" be "here"?
l 140-142: This text is repeated.
l 143-145: I would link this back to the common language to describe intraclass correlations, such as Shrout and Fleiss (1979) and/or Nakagawa and Schielzeth (2013).
Figure 3: Please flip the axes.
l 165: The word "do" is needed.
l 174: Why are capitals used here?
l 204-205: I would avoid saying "rose through the ranks".
l 206: A lesser extent to what?
l 213: What level of correlations is described as "substantial". Please refer to the language/cut-offs in Cohen (1992).
Figure 4B: I am afraid that I have an aversion to a correlation matrix presented in anything other than a table. It makes it easier to obtain those correlations from papers, too (you can copy and paste them).
l 235: I would just say the "smallest number" and not "minimal set".
l 416: Here I would say "each with a lid".
l 524: The acronym doesn't match the text it represents.
l 561: There is some formatting error here. Please correct it.
Thank you for asking me to review this manuscript. I hope the authors find my comments helpful.
Alexander Weiss [I sign my reviews]

References

- Cohen, J. (1992). A power primer. *Psychological Bulletin*, 112, 155-159.
<https://doi.org/10.1037/0033-2909.112.1.155>
Kaufman, A. B., Reynolds, M. R., & Kaufman, A. S. (2019). The structure of ape (hominoidea) intelligence. *Journal of Comparative Psychology*, 133, 92-105. <https://doi.org/10.1037/com0000136>
Nakagawa, S., & Schielzeth, H. (2013). rptR: Repeatability for Gaussian and non-Gaussian data <http://R-Forge.R-project.org/projects/rptr/>
Platt, J. R. (1964). Strong inference. *Science*, 146(3642), 347-353.
<https://doi.org/10.1126/science.146.3642.347>
Shrout, P. E., & Fleiss, J. L. (1979). Intraclass correlations: Uses in assessing rater reliability.

Reviewer #2 (Remarks to the Author):

This study presents novel, and much needed, findings on whether or not ape cognition research meets a number of implicit assumptions, namely: 1) group-level results are measured stably, 2) individual differences are measured reliably, and 3) cognitive performance is predictable. The authors test apes belonging to 4 species (chimpanzees, bonobos, gorillas, orangutans) in 5 cognitive tasks every two weeks for the duration of 14 data collection time points (Phase 1), followed by another 14 time points (Phase 2). They compare the findings in the two Phases. Overall, they showed that group-level performance (all species combined) was stable. Most tasks had high retesting correlations, and individual differences were mainly explained by stable differences between individuals, like group or age, and not by transient differences, like motivation or life events.

This study elegantly and thoroughly tests critical assumptions of comparative cognitive research with great apes. Overall, these findings will be of great interest to researchers across a number of areas, including anthropology, psychology, and evolutionary biology.

I have some small comments/suggestions below (specific comments). The only more substantial point is that I would like to see some more explicit discussion of the (lack of) species differences reported. The authors test for stability of group level performance where they combine all apes from the different species. Next, they look at the reliability of individual differences. It seems to me that it would make sense to also explicitly address the stability and reliability of group (incl. species) differences.

Specific comments:

L 110: Perhaps you can mention which 14 variables you refer to somewhere in the Intro?

L 112-113: Please add info on the overall length of the two phases.

Fig 1: Please briefly explain what Session 1 and Session 2 mean? Is there a break between two sessions? Are Session 1 and 2 done at each time point? And in both Phase 1 and 2? Please clarify. (I know I can find this in the suppl info, but it would be good to have the basic info in the main manuscript).

Why is 'Delay of gratification' only done in Phase 2? Please explain.

L 136-137: Are group level differences stable and measured reliably? I would like to see this addressed somewhere around here.

I see that the authors come back to this topic in the end of the Results section. But to me it seems like a question to be addressed following the question of 'Stability of group-level performance', and before looking at 'Reliability of individual differences'.

The finding that species differences are not reliably measured should could receive some more attention in Discussion.

L 165: ...we 'do' not exclusively...

L 306: 'median sample size = 7' not in superscript

Reviewer #3 (Remarks to the Author):

This paper presents the results of an impressive body of research investigating the reliability of measures used to evaluate 5 cognitive abilities in great apes, as well as how stable the individual differences are over time. Strikingly, most of the tests do not show a significant learning effect, which means multiple assays of the same ability are meaningful, and not polluted by learning effects. This is very useful information for those interested in longitudinal analyses. At the same time, there are reliable group effects and the time involved in research was a significant predictor. This should give pause to those comparing species when this factor is often a confounding variable. These ramifications are wide reaching, spanning research in behavioural ecology, comparative psychology, and evolutionary psychology.

Overall I find this to be a very valuable contribution to the literature, and something that has not been attempted at this scale before. As the authors point out – it should represent a new gold standard in establishing the validity of the methods used to assay cognitive skills.

The paper needs to be reviewed by a specialist statistician, as I am not expert enough to evaluate the specific methods used. From my understanding, the work is done carefully and thoroughly. The experiments are described in good detail, and run with appropriate controls.

My main comment therefore is to do with the clarity of the writing, and the inferences drawn from the different analyses. It seems very clear from the strong and significant correlations that the measures are reliable. And the SEM analyses, if I have followed it correctly, shows that the ranking of individuals

7on different experiments remains quite stable over repeated testing sessions. However, the inference drawn from this second fact – that the measures are therefore sensitive to traits such as the cognitive abilities being measured (such as causal inference) – rather than other traits, such as motivation, or attention – does not seem to me to be warranted. Why shouldn't motivational differences between individuals remain stable over time? Differences in sustained attention might also be an important predictor of ability, and not 'occasion specific'. And given the positive correlation between the difference assays (at least those in the physical domain) – it seems that the tests are not particularly specific to any one ability, but might rather task several abilities, with abilities that span different tests contributing to the variance in a substantial way. Motivation and attention would be reasonable candidates for such general differences between individuals. I would therefore recommend restricting the labelling of these different contributions toward the variance to something less value-laden (such as stable and occasion-specific variation) throughout the results, and then introduce a more nuanced discussion of what that might mean in the framing and discussion of the analysis.

A secondary comment concerns the relegation of one of the tasks to the supplementary materials. I think it is a pity not to include the discussion of this task in the main body of the paper. To me it serves a useful purpose, as it reveals what kind of profile a less meaningful test can show. If other researchers are to follow the lead set by this paper – and use repeated testing to investigate the psychometric profile of their tests before – for example- using them to compare species or cohorts - it would be useful to show both anchor points. However, the information is available in the supplementary material, so this is just a point of preference for the authors to consider.

In sum, I would like to recommend publication of this paper, following review by a statistical expert, and some consideration of the interpretation of the SEM analysis.

Reviewer #4 (Remarks to the Author):

The authors investigate various aspects of great ape cognition in a comparative study with 43 great apes from different species (Bonobos, Chimpanzees, Gorillas and Orangutans). In particular, they are interested in how stable, reliable, and predictable measures of great ape cognition are. And they explore their situation specificity and consistency and test the validity of the measures by considering correlations between cognitions. Overall, this is an interesting study that tries to advance the field methodologically. Consequently, a psychometric approach using structural equation modeling is applied. While I appreciate the authors attempt to apply state-of-the-art statistical methods in this context, I believe that there is some room for improvement, as I will outline below.

First, the authors claim that describing great ape cognition requires that "a) group-level results are measured stably, b) individual differences are measured reliably, and c) cognitive performance is predictable". These three assumptions are central and provide the structure for the entire manuscript. The authors take this for granted and don't provide a justification. In psychometrics, usually the three main quality criteria reliability, objectivity, and validity are considered along with their corresponding subdomains. While the authors' assumptions make sense intuitively and are related to the standard quality criteria, they are not identical and some aspects are missing.

8Second, I think it is unfortunate and somewhat inconsistent that the authors conducted fairly sophisticated statistical models and argue that such an approach is needed, but only report the results hidden in the online supplemental materials. In the main text, at least for testing the first two assumptions, they rely on oversimplified statistical estimates.

For investigating the first assumption of stable group-level results, they rely on simple means per time point and species, completely ignoring the time series nature of the measures and statistical uncertainty of the estimates. From visually inspecting the means for the first assumption, authors state that "group-level performance was largely stable or followed clear temporal patterns". This is a strong statement considering the figure and the means. It may be true for causal inference and to a lesser extent for quantity discrimination. But it is less clear for the other tasks. For example, Chimpanzee Group B has the lowest performance in the first period and rises to the top in the second. Without standard errors and statistical tests that account for the time series structure, it is really hard to tell. Simply relying on interpreting the lines in the figure in the main text of the manuscript seems not enough to me.

For testing the second assumption of reliable individual differences, the authors just report various lagged correlations, also ignoring autocorrelation, statistical uncertainty and measurement error. I don't think it makes much sense to investigate reliability of individual differences in this way. It is true that test-retest correlations are used in psychometrics as one option to assess reliability of measures. However, this procedure requires that the two time points are far enough apart that there are no learning or memory effects and so on (in technical terms, it requires that the two measures at different time points are parallel measures). This is simply not the case here. We probably have learning effects, memory effects and autoregressive effects and so on, so that the correlations are not useful for assessing reliability.

As mentioned above, I acknowledge that the authors conducted more appropriate analyses in the supplemental materials, but why not report these in the main text then? In my opinion, it should be the other way round. Additional descriptive statistics can be shown in supplemental materials but the proper analysis should appear in the main text.

Third, the group sizes are really low and this may be problematic for the statistical models and the generalizability of the results. I have seen the simulation study addressing sample size issues in the supplemental materials but this ignores the group sizes and just considers an overall sample size of $N=40$ (which is really low for SEM, probably only works with using prior information in a Bayesian framework and not in a Frequentist framework). All advanced analyses require that all individuals from all species are analyzed jointly, but there may be considerable differences between species. And the group comparisons are sometimes based on group sizes below five (e.g. for Orangutans at some time points). It is a lot of effort conducting such studies with great apes and I appreciate the time spend for this study, so my arguments should only be understood from a statistical point of view.

Further minor aspects:

- The LST models used in the main text are only described in the supplemental materials. Please briefly introduce the models that were estimated and the manifest indicators used with a couple of sentences in the main text and then refer the reader to the supplemental materials. Please explain which LST model was used (p.11 l. 172ff).
- Figure 2: Please add standard errors or confidence intervals for the means

- Figure 3 to me is counterintuitive and difficult to read. And it violates standard ways of displaying information. It seems as if a linear regression is depicted with "Distance btw. time points" as the dependent variable and Re-test correlation as the independent variable, which does not make sense conceptually. At least the axes should be interchanged. And I don't think the focus on time lags is particularly helpful. In essence, you just make an autocorrelation plot, that has been proposed by Box and Jenkins and is commonly used in time series analyses for model identification, here with interchanged axes. Also the distribution of correlations is difficult to interpret, because these are not independent observations. And why do you report the correlations in the upper left corner of the subfigures -- this is a really hard to interpret quantity. Finally, how did you compute the confidence bands? I guess confidence bands should not be shown (the default ggplot2 settings assume that the points are observations not correlations)?

Some terminological difficulties that I experienced while reading the article:

- How is reliability of individual differences defined? How is stability defined. What do you mean with structure of individual differences. These terms have a clear meaning in psychometrics, but this is sometimes not in line with the way the authors use these terms.
- E.g., the term stability is used with different meanings (temporal stability, stability across repeated studies and stability in group differences).
- Or "with structure we not exclusively mean the relations between different cognitive tasks..." (p.10, l.165) was confusing to me.
- When you apply LST theory, the more technical terms such as traits, state residuals, consistency, situation/occasion specificity, and variance of latent state variables are introduced but not always used. This makes it more difficult to read.
- It is generally not correct to say that "In the LSTM context, reliability is estimated as split-half reliability based on repeated parallel measurements per time point"
- "Correlation reliably different from zero" is a strange formulation

Typos:

- p.7, l. 120. Should this be "from each species"?
- p.9, l. 140-142. I guess these lines should be deleted?
- Sometimes "latent state-trait models (LSTM)" is used and sometimes LST models is used as abbreviation. Please make consistent. I would prefer the latter.

*****END*****

Author Rebuttal, first revision:

10Reviewer #1

This paper describes a study of 43 great apes. The authors tested whether several cognitive traits, each measured on multiple occasions, in these apes was stable over time at the level of the group, whether individual differences in these traits were reliable, and whether they were stable within the individual/unaffected by internal factors. What I appreciated about this study was that the authors did a nice job of choosing data analytic approaches that were suitable for answering the questions that they sought to address. Finally, the writing, and explanations of their methods, was clear for the most part. All that said there were some aspects of this manuscript that I thought could use work.

Response: Thank you very much for the overall positive assessment of our work and the helpful comments.

First, to my mind, this paper seemed more method-oriented than problem-oriented (see, e.g., Platt, 1964). In other words, I think the authors would be much better off focusing on what questions about cognition in these species they wish to address and what different findings would mean in this regard. I am not sure how much revising this will take, but it will require some thought.

Response: Thank you for this comment. We rewrote and restructured most of the introduction to make the “problem” we are addressing more clear. In short, we test fundamental assumptions about the nature and structure of cognitive abilities of great apes that usually go untested. In the empirical part of the project, we go through some of the most widely studied aspects of cognition to provide such evidence. We hope our edits make this focus clearer.

Continuing, although I cannot speak for the authors, I was under the impression that what they had in mind is that studies of cognition in primates rested on several assumptions, but that these assumptions

were not tested, and that the goal of their study was to test these assumptions. If so, that is fine provided others had not tested these assumptions before or there were still gaps in this literature. Related to this, one key paper that the authors did not cite was that by Kaufman et al. (2019). The authors may also wish to extend their literature review to go beyond great apes. I don't think every last paper needs to be cited, but more could be if they wish to make the case for this study.

Response: Thank you very much for pointing us to this paper which we must have overlooked. We now cite it in the introduction on page 6 and we also added a new section to the introduction in which we discuss work beyond great apes on page 6.

Second, I failed to grasp the importance of the analyses/question about the "stability of group-level performance". How is it critical whether one species or group or another shows the same age trajectories (or sex differences, etc.) than others, or even a clear pattern, for this kind of trait? Many traits change over time, much in the same way as physical characteristics, although some do not, but that does not make one more or less trait-like. If there is some expectation, grounded in trait theory and cognitive psychology, of why one should see the same pattern of change across groups or species, then the authors did not clearly spell out why one should have these expectations.

Response: We apologise for this confusion. The term "group-level" stability was not well chosen and apparently caused confusion (see also reviewer 3). What we meant by this was that the overall results should be replicable. Usually, when we run a study in comparative psychology to investigate if a given species or clade has ability X, we take the result we obtain in one sample to be representative, that is, we assume that if we were to run the study again, we would get the same or a similar result. Here we tested this assumption by not only repeating the same study twice but a total of 28 times. Our goal was not to say that differences between groups of animals (e.g. chimpanzee B group) need to be stable or that such groups should change in the same way but that the aggregated performance of a given sample in a task should be more or less stable if we think that it is produced by a certain cognitive ability. To avoid this confusion, we changed the wording to "task-level" robustness.

Third, the most important finding was at the end of the Result section (275-280). The best evidence that these are traits or trait-like is that most of the variance was not due to transient, situational factors, but consistent between-individual variance. This needs more prominence.

Response: We are glad to see that you found these results convincing. We expanded this section and also highlighted this aspect of the study more in the introduction section.

Fourth, I disagree that their study addresses the question of whether the results of their study and that of others can be understood using associative learning (see paragraph starting on line 350). The second statement (lines 352-353) is simply an assertion with nothing backing it. Just because the same experimented was present, differential reinforcement was used, etc., does not mean that a steady increase, let alone a uniform increase across individuals, should occur. Here, again, the paper by Platt is an outstanding guide to working out what hypotheses/explanations your study can and cannot rule out. I would omit this text.

Response: We take the point that our study did not explicitly address the question of the validity of an associative learning interpretation of our results, which is why we only raise this in the discussion and not the results section. However, to make our stance more clear, we changed the wording in the alleged paragraph and eliminated the claim that associative learning can be ruled out given the non-existing learning curve in our data. Furthermore, we now explicitly invite others to use our data to test potential associative learning accounts that would give rise to the patterns we observed (or not). We hope that these changes alleviate your concerns with the paragraph because we would be reluctant to take it out since another reviewer explicitly highlighted that they liked it.

In addition to these major points, I noticed some minor matters that the authors should address. I will refer to each by line numbers.

ll 31-32: The sentence here is awkward and should be rewritten.

l 33: The use of the word "licensed" here is odd.

Response: We changed this sentence and removed the word "licensed"

ll 33-34: This is not clear.

Response: We changed the sentence. Thank you.

l 35: It is probably best to refer to "relationships" in this sentence.

Response: We changed the wording throughout the paper.

l 53: I would omit the phrase "in time"

Response: We removed the phrase. Thank you.

l 53: The comparative method could be brought up sooner. A brief mention and citation of the other approaches that you refer to with relevant references is all that is necessary.

Response: We would prefer to keep the opening of the introduction as it is. Given the interdisciplinary readership of the journal, we think it helps to familiarise the reader with the problem we are studying.

l 55: The word "backward" is needed here.

Response: In our version of the paper, it was there - anyway, we made sure it is included in the revision.

l 58: Do not hyphenate "nonhuman". Also, change "such as" to "this".

Response: We changed it, thank you.

ll 72-73: I would rephrase this to avoid the word "prototypical".

Response: We changed it, thank you.

l 76: References are needed for the end of this sentence.

Response: In our version of the paper there were references there. We made sure that they are also included in the revision. However, we also changed this section of the paper and this sentence, so it is no longer the same.

l 78: I think it would be better to refer to "assumptions" and stick to that. Something other than "problems" would be advisable in any event, and please be consistent with the terminology throughout, i.e., edits to lines 92 and 93, and so on.

Response: When rewriting the introduction we removed this sentence. We also made sure to be consistent with our terminology throughout.

l 100: Should "below" be "here"?

Response: We changed it, thank you.

ll 140-142: This text is repeated.

Response: We removed the repeated part.

ll 143-145: I would link this back to the common language to describe intraclass correlations, such as Shrout and Fleiss (1979) and/or Nakagawa and Schielzeth (2013).

Response: We think this recommendation is based on our unlucky use of the word "group-level". We changed this in the revision so that we think this no longer applies.

Figure 3: Please flip the axes.

Response: We changed Figure 3 (now Figure 4) completely which included flipping the axis.

l 165: The word "do" is needed.

Response: Added it, thank you!

l 174: Why are capitals used here?

Response: We changed it, thank you.

ll 204-205: I would avoid saying "rose through the ranks".

Response: We changed the wording, thank you.

I 206: A lesser extent to what?

Response: We changed the wording to make this more clear.

I 213: What level of correlations is described as "substantial". Please refer to the language/cut-offs in Cohen (1992).

Response: We defined substantial correlations as correlations with 95% CrIs not including zero. We added this information to the text and also added a reference to Cohen on page 15. We apologise for being unclear here.

Figure 4B: I am afraid that I have an aversion to a correlation matrix presented in anything other than a table. It makes it easier to obtain those correlations from papers, too (you can copy and paste them).

Response: We completely understand the general preference for correlations to be presented in a table. However, in this case, we think the figure is better suited because it makes it easier to deal with the empty cells and the two correlations in the case of inference by exclusion in Phase 2. Given that the data is publicly available, the correlations are also easily accessible.

I 235: I would just say the "smallest number" and not "minimal set".

Response: We changed it in line with the suggestion. Thank you.

I 416: Here I would say "each with a lid".

Response: We changed it in line with the suggestion. Thank you.

I 524: The acronym doesn't match the text it represents.

Response: We changed it. Thank you.

I 561: There is some formatting error here. Please correct it.

Response: We corrected it, thank you.

*Thank you for asking me to review this manuscript. I hope the authors find my comments helpful.
Alexander Weiss [I sign my reviews]*

References

- Cohen, J. (1992). A power primer. *Psychological Bulletin*, 112, 155-159. <https://doi.org/10.1037/0033-2909.112.1.155>
- Kaufman, A. B., Reynolds, M. R., & Kaufman, A. S. (2019). The structure of ape (homoidea) intelligence. *Journal of Comparative Psychology*, 133, 92-105. <https://doi.org/10.1037/com0000136>
- Nakagawa, S., & Schielzeth, H. (2013). rptR: Repeatability for Gaussian and non-Gaussian data <http://R-Forge.R-project.org/projects/rptr/>
- Platt, J. R. (1964). Strong inference. *Science*, 146(3642), 347-353. <https://doi.org/10.1126/science.146.3642.347>
- Shrout, P. E., & Fleiss, J. L. (1979). Intraclass correlations: Uses in assessing rater reliability. *Psychological Bulletin*, 86(2), 420-428. <https://doi.org/10.1037/0033-2909.86.2.420>

Reviewer #2

This study presents novel, and much needed, findings on whether or not ape cognition research meets a number of implicit assumptions, namely: 1) group-level results are measured stably, 2) individual differences are measured reliably, and 3) cognitive performance is predictable. The authors test apes belonging to 4 species (chimpanzees, bonobos, gorillas, orangutans) in 5 cognitive tasks every two weeks for the duration of 14 data collection time points (Phase 1), followed by another 14 time points (Phase 2). They compare the findings in the two Phases. Overall, they showed that group-level performance (all species combined) was stable. Most tasks had high retesting correlations, and individual differences were mainly explained by stable differences between individuals, like group or age, and not by transient differences, like motivation or life events.

This study elegantly and thoroughly tests critical assumptions of comparative cognitive research with great apes. Overall, these findings will be of great interest to researchers across a number of areas, including anthropology, psychology, and evolutionary biology.

Response: Thank you very much for the positive assessment of our work. We are glad you think it will be useful.

I have some small comments/suggestions below (specific comments). The only more substantial point is that I would like to see some more explicit discussion of the (lack of) species differences reported. The authors test for stability of group level performance where they combine all apes from the different species. Next, they look at the reliability of individual differences. It seems to me that it would make sense to also explicitly address the stability and reliability of group (incl. species) differences.

Response: Thank you for this comment. We completely agree that it would be very interesting to further explore species differences. The main reason why we are hesitant to do so is the massive difference in sample size between species and also species by time point combinations. For example, while we have more than 20 chimpanzees that participate on a regular basis, we only have around five gorillas. As a consequence, the species-level estimates will differ tremendously in precision, and we will not be able to distinguish individual-level from species-level differences. Nevertheless, we hope that this study lays the

groundwork for future collaborative work in which we will get more comparable sample sizes for each species. In the present analysis, we only look at group differences when predicting cognitive performance (section Predictability of individual differences). In this case, however, the interpretation rests on relative differences between groups and not on the numerical estimates for each group. In sum and in line with the comment, we have expanded the discussion section dedicated to discussing species differences and also mention the problems associated with comparing species based on the current study (page 20).

Specific comments:

L 110: Perhaps you can mention which 14 variables you refer to somewhere in the Intro?

Response: We added the variables at the end of the introduction on pages 7 - 8.

L 112-113: Please add info on the overall length of the two phases.

Response: We added the total duration on page 8. The exact dates for each phase can be found in the methods section on page 26.

Fig 1: Please briefly explain what Session 1 and Session 2 mean? Is there a break between two sessions? Are Session 1 and 2 done at each time point? And in both Phase 1 and 2? Please clarify. (I know I can find this in the suppl info, but it would be good to have the basic info in the main manuscript).

Response: Thank you, we added this information to the figure legend.

Why is 'Delay of gratification' only done in Phase 2? Please explain.

Response: In Phase 1, we had a different task for executive functions, the switching task described in the supplementary material. Because this task did not yield stable task- or individual-level results, we decided to replace it with a different task in Phase 2. We added this information to the introduction on page 7.

L 136-137: Are group level differences stable and measured reliably? I would like to see this addressed somewhere around here.

I see that the authors come back to this topic in the end of the Results section. But to me it seems like a question to be addressed following the question of 'Stability of group-level performance', and before looking at 'Reliability of individual differences'.

Response: We apologize for the confusing way in which we used the term "group level". We have rewritten this section of the results to avoid confusion. We hope the new version is now clear. In short, if "group-level" is taken to refer to groups of animals (e.g. chimpanzee A group), we do not address this question due to the reasons mentioned above (differences in sample sizes and low sample sizes for species by time-point combinations).

The finding that species differences are not reliably measured should could receive some more attention in Discussion.

Response: We expanded the discussion on page 20. Thank you.

L 165: ...we 'do' not exclusively...

Response: Thank you, we corrected it.

L 306: 'median sample size = 7' not in superscript

Response: Thank you, we corrected it.

Reviewer #3

This paper presents the results of an impressive body of research investigating the reliability of measures used to evaluate 5 cognitive abilities in great apes, as well as how stable the individual differences are over time. Strikingly, most of the tests do not show a significant learning effect, which means multiple assays of the same ability are meaningful, and not polluted by learning effects. This is very useful information for those interested in longitudinal analyses. At the same time, there are reliable group effects and the time involved in research was a significant predictor. This should give pause to those comparing species when this factor is often a confounding variable. These ramifications are wide reaching, spanning research in behavioural ecology, comparative psychology, and evolutionary psychology.

Overall I find this to be a very valuable contribution to the literature, and something that has not been attempted at this scale before. As the authors point out – it should represent a new gold standard in establishing the validity of the methods used to assay cognitive skills.

The paper needs to be reviewed by a specialist statistician, as I am not expert enough to evaluate the specific methods used. From my understanding, the work is done carefully and thoroughly. The experiments are described in good detail, and run with appropriate controls.

Response: Thank you very much for the very encouraging assessment of our work.

My main comment therefore is to do with the clarity of the writing, and the inferences drawn from the different analyses. It seems very clear from the strong and significant correlations that the measures are reliable. And the SEM analyses, if I have followed it correctly, shows that the ranking of individuals on different experiments remains quite stable over repeated testing sessions. However, the inference drawn from this second fact – that the measures are therefore sensitive to traits such as the cognitive abilities being measured (such as causal inference) – rather than other traits, such as motivation, or attention – does not seem to me to be warranted. Why shouldn't motivational differences between individuals remain stable over time? Differences in sustained attention might also be an important predictor of ability, and not 'occasion specific'. And given the positive correlation between the difference assays (at least those in

the physical domain) it seems that the tests are not particularly specific to any one ability, but might rather task several abilities, with abilities that span different tests contributing to the variance in a substantial way. Motivation and attention would be reasonable candidates for such general differences between individuals. I would therefore recommend restricting the labelling of these different contributions toward the variance to something less value-laden (such as stable and occasion-specific variation) throughout the results, and then introduce a more nuanced discussion of what that might mean in the framing and discussion of the analysis.

Response: Thank you very much for this comment. We completely agree that motivation and attention could also systematically and stably differ between individuals and that our analysis does not address the question of whether this is the case. We also very much like the suggestion that such differences in domain-general abilities like attention or attentiveness are responsible for the trait correlations we found. In line with this, we have adjusted the text in several places (pages 13 and 21). Because we make it clear what our analysis can – and cannot – before we discuss the results, we would prefer to keep the (less technical) terminology. We think that makes it easier to follow the results, especially for readers less familiar with these analytical methods.

A secondary comment concerns the relegation of one of the tasks to the supplementary materials. I think it is a pity not to include the discussion of this task in the main body of the paper. To me it serves a useful purpose, as it reveals what kind of profile a less meaningful test can show. If other researchers are to follow the lead set by this paper – and use repeated testing to investigate the psychometric profile of their tests before – for example- using them to compare species or cohorts - it would be useful to show both anchor points. However, the information is available in the supplementary material, so this is just a point of preference for the authors to consider.

Response: Thank you for this suggestion – we very much struggled with the question of whether or not to include the task in the main text. We agree that it would serve a very useful purpose. However, the main reason why we left it out was that we could not have applied the same analytical methods to it. The dependent variable in this task was computed as a ratio between two phases. As such, there is only a single data point per participant and time point. This would not have allowed us to use LS(T)Ms to analyse it because we could not have estimated the latent variables (and the reliability) based on two parallel test halves per time point. We hope that makes our decision more clear. Thank you for your understanding.

In sum, I would like to recommend publication of this paper, following review by a statistical expert, and some consideration of the interpretation of the SEM analysis.

Response: Thank you, we hope we successfully addressed all your concerns.

Reviewer #4

The authors investigate various aspects of great ape cognition in a comparative study with 43 great apes from different species (Bonobos, Chimpanzees, Gorillas and Orangutans). In particular, they are interested in how stable, reliable, and predictable measures of great ape cognition are. And they explore

their situation specificity and consistency and test the validity of the measures by considering correlations between cognitions. Overall, this is an interesting study that tries to advance the field methodologically. Consequently, a psychometric approach using structural equation modeling is applied. While I appreciate the authors attempt to apply state-of-the-art statistical methods in this context, I believe that there is some room for improvement, as I will outline below.

First, the authors claim that describing great ape cognition requires that "a) group-level results are measured stably, b) individual differences are measured reliably, and c) cognitive performance is predictable". These three assumptions are central and provide the structure for the entire manuscript. The authors take this for granted and don't provide a justification. In psychometrics, usually the three main quality criteria reliability, objectivity, and validity are considered along with their corresponding subdomains. While the authors' assumptions make sense intuitively and are related to the standard quality criteria, they are not identical and some aspects are missing.

Response: Thank you for this comment and for linking our work more closely to the psychometric literature. The reason we did not structure our manuscript along the three main quality criteria is that our goal was not necessarily to construct a new test or task to measure specific cognitive abilities but rather to use already existing tasks to evaluate the common practices in the field of comparative cognition. Our target audience is researchers working on comparative cognition, and we therefore decided to structure the manuscript along categories (or questions) that are immediately relevant to their everyday work. We agree that these implicitly relate to the three quality criteria reliability, objectivity and validity, but they refer less to specific tasks than to more general research practice.

With respect to the three quality criteria, we can say the following about the tasks we used: Most of the tasks measured the underlying ability in a reliable way. That is, assumption b) directly relates to the reliability criterion, which we investigated in the SEM analyses. With respect to objectivity, we assessed inter-rater agreement for the coding of apes' choices and behaviors and found solid agreement between raters in all tasks. With respect to the different aspects of validity, we may say that the tasks have content validity because experts in the field regularly use them to assess said abilities. For construct validity, we can look at the correlations between traits. The tasks were selected to measure different cognitive abilities, and thus we did not expect very high correlations between tasks. On the other hand, we expect there to be shared processes between tasks (to be explicated and formalized in formal models), so we did not expect correlations to be 0. The pattern of correlations we found could be interpreted as supporting these expectations; however, we have to say that the study was not designed to assess convergent or discriminant validity of the tasks involved.

We have restructured the introduction of the manuscript to provide a clearer justification for the way in which we structure the manuscript.

Second, I think it is unfortunate and somewhat inconsistent that the authors conducted fairly sophisticated statistical models and argue that such an approach is needed, but only report the results hidden in the online supplemental materials. In the main text, at least for testing the first two assumptions, they rely on oversimplified statistical estimates.

For investigating the first assumption of stable group-level results, they rely on simple means per time point and species, completely ignoring the time series nature of the measures and statistical uncertainty of the estimates. From visually inspecting the means for the first assumption, authors state that "group-level performance was largely stable or followed clear temporal patterns". This is a strong statement considering the figure and the means. It may be true for causal inference and to a lesser extent for quantity discrimination. But it is less clear for the other tasks. For example, Chimpanzee Group B has the lowest performance in the first period and rises to the top in the second. Without standard errors and statistical tests that account for the time series structure, it is really hard to tell. Simply relying on interpreting the lines in the figure in the main text of the manuscript seems not enough to me.

Response: Thank you for this comment. As we mentioned above, we used “group-level” here not to talk about a group of participants (e.g. chimpanzee B group) but about the aggregated performance across all participants. That is, with the term “group-level performance” we did not refer to differences in the trajectories between different species. We agree that we cannot draw conclusions with respect to (the stability of) these differences simply by visual inspection of the figure. However, as outlined below, the sample sizes are not sufficient to investigate species by time-point differences by use of statistical tests. We have now changed the term to “task-level” performance to avoid confusion. We apologize for this confusing choice of words. In addition, we very much appreciate the more general argument that the results reported fall short compared to the ones reported in the supplementary material. As a consequence, we have moved some of them to the main text.

For testing the second assumption of reliable individual differences, the authors just report various lagged correlations, also ignoring autocorrelation, statistical uncertainty and measurement error. I don't think it makes much sense to investigate reliability of individual differences in this way. It is true that test-retest correlations are used in psychometrics as one option to assess reliability of measures. However, this procedure requires that the two time points are far enough apart that there are no learning or memory effects and so on (in technical terms, it requires that the two measures at different time points are parallel measures). This is simply not the case here. We probably have learning effects, memory effects and autoregressive effects and so on, so that the correlations are not useful for assessing reliability. As mentioned above, I acknowledge that the authors conducted more appropriate analyses in the supplemental materials, but why not report these in the main text then? In my opinion, it should be the other way round. Additional descriptive statistics can be shown in supplemental materials but the proper analysis should appear in the main text.

Response: Thank you very much for this comment. We changed this section of the results and the analysis. We mention the lagged correlations only in a purely descriptive context and then analyse the reliability of the tasks based on the methods formerly described in the supplementary material (LS(T) models).

Third, the group sizes are really low and this may be problematic for the statistical models and the generalizability of the results. I have seen the simulation study addressing sample size issues in the supplemental materials but this ignores the group sizes and just considers an overall sample size of $N=40$ (which is really low for SEM, probably only works with using prior information in a Bayesian framework and not in a Frequentist framework). All advanced analyses require that all individuals from all species

are analyzed jointly, but there may be considerable differences between species. And the group comparisons are sometimes based on group sizes below five (e.g. for Orangutans at some time points). It is a lot of effort conducting such studies with great apes and I appreciate the time spend for this study, so my arguments should only be understood from a statistical point of view.

Response: We agree that – from a statistical point of view – the sample size is small. Furthermore, as mentioned above, we would also be very much interested in exploring species differences. However, collecting data from larger samples is very difficult with great apes and primates in general (a recent review found the median sample size across studies to be 7).

As comparisons between species were not the aim of the study and species by time point sample sizes are very low we refrain from interpreting the purely descriptive differences between species displayed in the figure. We rephrased respective sentences to avoid confusion. Regarding the simulation study, we do now stress that the results do only apply to estimation in a Bayesian framework and are not meant to provide general recommendations, but only serve the purpose to check whether results in the present study are to be trusted.

Nevertheless, we think our study strikes a good balance between statistical sophistication and level of detail. As such, we think that in addition to providing interesting insights, it can make an important contribution to overcoming the issue of small sample sizes by providing a solid empirical basis for future large-scale collaborations to build on.

Further minor aspects:

- The LST models used in the main text are only described in the supplemental materials. Please briefly introduce the models that were estimated and the manifest indicators used with a couple of sentences in the main text and then refer the reader to the supplemental materials. Please explain which LST model was used (p.11 l. 172ff).

Response: We added more information about the models in the results and the methods section and referred the reader to the methods section and the supplementary material for details.

- Figure 2: Please add standard errors or confidence intervals for the means

Response: We think this comment is based on the misunderstanding of our use of the term “group-level”. Since we only interpreted performance across all participants, we only added confidence intervals for the mean across all participants. Adding species or group-specific confidence intervals would make the graph difficult to read and might tempt the reader to interpret group/species differences in a way we would like to avoid.

- Figure 3 to me is counterintuitive and difficult to read. And it violates standard ways of displaying information. It seems as if a linear regression is depicted with "Distance btw. time points" as the dependent variable and Re-test correlation as the independent variable, which does not make sense conceptually. At least the axes should be interchanged. And I don't think the focus on time lags is particularly helpful. In essence, you just make a autocorrelation plot, that has been proposed by Box and

Jenkins and is commonly used in time series analyses for model identification, here with interchanged axes. Also the distribution of correlations is difficult to interpret, because these are not independent observations. And why do you report the correlations in the upper left corner of the subfigures -- this is a really hard to interpret quantity. Finally, how did you compute the confidence bands? I guess confidence bands should not be shown (the default ggplot2 settings assume that the points are observations not correlations)?

Response: Thank you very much for this comment. We now see that the way we visualized the data here was inappropriate, and we completely changed the figure. That is, we flipped the axis and removed the confidence bands and the correlation coefficients. As such, the figure now only serves a descriptive purpose, namely to illustrate that correlations decrease with time distance. As mentioned above, in the text, we make clear that these correlations are difficult to interpret and then describe more appropriate statistical analyses.

Some terminological difficulties that I experienced while reading the article:

- How is reliability of individual differences defined? How is stability defined. What do you mean with structure of individual differences. These terms have a clear meaning in psychometrics, but this is sometimes not in line with the way the authors use these terms.

- E.g., the term stability is used with different meanings (temporal stability, stability across repeated studies and stability in group differences).

- Or "with structure we not exclusively mean the relations between different cognitive tasks..." (p.10, l.165) was confusing to me.

Response: We went over the manuscript again and made sure to provide proper definitions of these terms whenever we introduce them. We now only use the term stability when referring to temporal stability of inter-individual differences. Thank you for pointing this out to us.

- When you apply LST theory, the more technical terms such as traits, state residuals, consistency, situation/occasion specificity, and variance of latent state variables are introduced but not always used. This makes it more difficult to read.

Response: We tried to "translate" these technical terms into concepts that are more familiar to readers with a comparative psychology or evolutionary anthropology background. We now make use of these terms wherever we can without rendering the text incomprehensible for our main readership. Given the broad and diverse readership of the journal, we hope this improves the readability of the paper for non-experts.

- It is generally not correct to say that "In the LSTM context, reliability is estimated as split-half reliability based on repeated parallel measurements per time point"

Response: We apologize, this sentence referred to our specific application only. We removed this sentence.

- *"Correlation reliably different from zero" is a strange formulation*

Response: We changed this formulation; we meant correlations that have Credible Intervals not overlapping with zero.

Typos:

- p.7, l. 120. *Should this be "from each species"?*

Response: No, we aggregated performance across all species (see discussion points above on the confusion around our use of the term "group-level").

- p.9, l. 140-142. *I guess these lines should be deleted?*

Response: Yes, we apologize for not spotting this. We removed them.

- *Sometimes "latent state-trait models (LSTM)" is used and sometimes LST models is used as abbreviation. Please make consistent. I would prefer the latter.*

Response: Thank you, we changed it in line with your suggestion.

Decision Letter, second revision:

28th February 2023

Dear Dr. Bohn,

Thank you for submitting your revised manuscript "Probing the structure, stability, and predictability of great ape cognition" (NATECOLEVOL-220817321B). It has now been seen again by the original reviewers and their comments are below. The reviewers find that the paper has improved in revision, and therefore we'll be happy in principle to publish it in Nature Ecology & Evolution, pending minor revisions to satisfy the reviewers' final requests and to comply with our editorial and formatting guidelines.

[REDACTED]

Reviewer #1 (Remarks to the Author):

Review of MS NATECOLEVOL-220817321B: Probing the structure, stability, and predictability of great ape cognition

This is a revision of a manuscript. The authors tested whether several cognitive traits, each measured on multiple occasions, in 43 apes was stable over time at the level of the group, whether individual differences in these traits were reliable, and whether they were stable within the individual/unaffected by internal factors. As before, I appreciate the authors' choice of data analytic approaches, and I thought the writing, and explanations of their methods, were largely clear.

My main concerns were as follows. First, the paper was method-oriented rather than problem-oriented, that is, they did not focus on the main research questions in the Introduction. Second, I had

25not understood what they meant about “stability of group-level performance” and why my understanding of what they meant was important. Third, I disagreed that their study addressed the question of whether the results of their study (and others) can be understood using associative learning. Finally, I had suggested several minor edits that they needed to address, including problems with figures.

With respect to my first concern, the Introduction has been improved, so maybe it’s just me—and if it is, I apologize—but the it still seems more methods- than problem-oriented. I would just like the authors to present succinctly some statement about what possible explanation their study will exclude. It would be nice to see how they got to this question, too.

With respect to my second concern, the authors did a great job addressing it in the letter. I now see what they mean, although I am not certain that framing what they did as a replication (or something like it), as I seem to think they did (lines 162-163), is correct. Replication concerns repeating the study in new data (see <https://doi.org/10.1371/journal.pbio.3000691>).

I also still think there was too much on associative learning/simpler processes (my third concern) in the Discussion section (lines 438-450). It has been a while since I dug into the learning literature, but just because a response does not steadily increase or decreases over time does not mean that there is no simpler process. For instance, the authors cannot exclude the possibility that the decrease in gaze-following (lines 445-446) is an instance in which the animals learn to ignore a cue that is not ecologically valid *sensu* Tolman and Brunswick (<https://doi.org/10.1037/h0062156>). Related to this, latent learning (<https://doi.org/10.1037/h0062733>) may explain the findings reported on lines 446-447. As I stated before, the current study simply cannot exclude various alternatives and so the authors are best off not discussing the matter. Note that I realize my interpretation clashes with that of Reviewer 3, but I cannot see how these data speak to this question.

In addition to highlighting these matters, and some minor issues, I wanted to turn to how the authors addressed an important point raised by Reviewer 3 regarding other traits, for example those related to motivation. One place in which I presume the authors addressed this was in lines 301-303. Data from several studies on the association between personality traits and cognitive performance simply do not support the nomological net that the authors are trying to cast (off the top of my head, <https://doi.org/10.1098/rsos.170169>; <https://doi.org/10.12966/abc.02.04.2016>; <https://doi.org/10.7717/peerj.9707>; <https://doi.org/10.1037/a0031723>; <https://doi.org/10.1007/s10071-013-0603-5>).

My additional minor comments follow and I refer to the line numbers in the manuscript.

line 28: I would omit “uniquely” the phrase “uniquely human cognition” has become a cliché, in all honesty, and it seems redundant anyway.

line 29: I would delete “specific”.

line 42: The use of “structured” here is awkward.

line 43: I think the word “abiding” can be omitted.

line 51: I’d move “directly” so that it follows “hominins” or delete it.

line 54: Insert “also” in this sentence and delete “now-”.

line 63: Delete the hyphen between “non” and “human”.

line 65: The word "several" here is redundant.
line 68: I think it's "researcher degrees of freedom".
line 71: Please omit the parentheses.
line 76: I'd say "the behavior" and omit "which is the basic material".
line 78: A reference is needed for the statement ending on this line.
lines 81-82: The text "asking these ... asking questions" is awkward, probably because "asking" is used twice here.
line 93: The phrase "in order to" can (and should) be written "to".
lines 94-95: The sentence spanning these lines repeats earlier text (lines 81-82), so consider revising some of the text on this page.
line 105: The phrase "undertook notable effort" is awkward.
lines 113-114: "Based on a..." should be "Using a..." You should also cite the Campbell and Fiske's 1959 paper from Psychological Bulletin here.
line 120: The phrase "general cognitive ability" shouldn't be in single quotes.
lines 122-123: I am not sure what this sentence is doing here. I would delete it.
line 124: Delete "seminal" please.
line 132: Delete "the nature of" please.
line 134: I think "directly" can be deleted.
lines 135-136: The text ending the sentence is awkward.
line 138: The word "across" can be deleted.
lines 142-145: This text is very awkward. It needs revising.
line 167: I think "percent correct" should be inserted after "mean".
line 168: "Structural Equation Modeling" should be in sentence case, as should "Latent State models", and similar.
line 179: Should it read "performance was"?
line 192: Instead of "licensed", use "supported", "were consistent with", or similar.
line 198: Insert "(total)" after "observed".
line 200: I would delete "(theoretically)".
line 207: I think another em dash is needed.
line 212: I think "two" can be deleted.
line 214: I would split the sentence at the end of this line.
line 219: A "full" what?
line 226: What "stands out"?
line 242: Fluctuations in what?
line 244: The phrase "tease them apart" is a cliché and has been used before here, I think, so change it, please.
line 251: The text on this line is awkward.
lines 346-347: This sentence is awkward.
line 391: Change "vs." to "and".
lines 470-471: The names of the species should not be capitalized. Also, should the species scientific names be noted?
line 475: I would delete the word "groups" within the parentheses and also the word "strictly".
lines 483-487: I think this text belongs under "Procedure" and the "Material" heading can be omitted.

Thank you for asking me to review this manuscript. I hope the authors find my comments helpful.

27Alexander Weiss [I sign my reviews]

Reviewer #4 (Remarks to the Author):

I appreciate the revision. The authors have addressed most of my points from the previous round and I have no further comments.

Our ref: NATECOLEVOL-220817321B

8th March 2023

Dear Dr. Bohn,

Thank you for your patience as we've prepared the guidelines for final submission of your Nature Ecology & Evolution manuscript, "Probing the structure, stability, and predictability of great ape cognition" (NATECOLEVOL-220817321B). Please carefully follow the step-by-step instructions provided in the attached file, and add a response in each row of the table to indicate the changes that you have made. Please also check and comment on any additional marked-up edits we have proposed within the text. Ensuring that each point is addressed will help to ensure that your revised manuscript can be swiftly handed over to our production team.

****We would like to start working on your revised paper, with all of the requested files and forms, as soon as possible (preferably within two weeks). Please get in contact with us immediately if you anticipate it taking more than two weeks to submit these revised files.****

28In recognition of the time and expertise our reviewers provide to Nature Ecology & Evolution's editorial process, we would like to formally acknowledge their contribution to the external peer review of your manuscript entitled "Probing the structure, stability, and predictability of great ape cognition". For those reviewers who give their assent, we will be publishing their names alongside the published article.

Nature Ecology & Evolution offers a Transparent Peer Review option for new original research manuscripts submitted after December 1st, 2019. As part of this initiative, we encourage our authors to support increased transparency into the peer review process by agreeing to have the reviewer comments, author rebuttal letters, and editorial decision letters published as a Supplementary item. When you submit your final files please clearly state in your cover letter whether or not you would like to participate in this initiative. Please note that failure to state your preference will result in delays in accepting your manuscript for publication.

Cover suggestions

As you prepare your final files we encourage you to consider whether you have any images or illustrations that may be appropriate for use on the cover of Nature Ecology & Evolution.

Nature Ecology & Evolution has now transitioned to a unified Rights Collection system which will allow our Author Services team to quickly and easily collect the rights and permissions required to publish your work. Approximately 10 days after your paper is formally accepted, you will receive an email in providing you with a link to complete the grant of rights. If your paper is eligible for Open Access, our Author Services team will also be in touch regarding any additional information that may be required to arrange payment for your article.

Please note that Nature Ecology & Evolution is a Transformative Journal (TJ). Authors may publish their research with us through the traditional subscription access route or make their paper immediately open access through payment of an article-processing charge (APC). Authors will not be required to make a final decision about access to their article until it has been accepted.

[href="https://www.springernature.com/gp/open-research/transformative-journals"](https://www.springernature.com/gp/open-research/transformative-journals) Find out more about Transformative Journals

Authors may need to take specific actions to achieve [compliance](https://www.springernature.com/gp/open-research/funding/policy-compliance-faqs) with funder and institutional open access mandates. If your research is supported by a funder that requires immediate open access (e.g. according to [Plan S principles](https://www.springernature.com/gp/open-research/plan-s-compliance)) then you should select the gold OA route, and we will direct you to the compliant route where possible. For authors selecting the subscription publication route, the journal's standard licensing terms will need to be accepted, including [self-archiving-and-license-to-publish](https://www.nature.com/nature-portfolio/editorial-policies/self-archiving-and-license-to-publish). Those licensing terms will supersede any other terms that the author or any third party may assert apply to any version of the manuscript.

[REDACTED]

[REDACTED]

Reviewer #1:

Remarks to the Author:

Review of MS NATECOLEVOL-220817321B: Probing the structure, stability, and predictability of great ape cognition

This is a revision of a manuscript. The authors tested whether several cognitive traits, each measured on multiple occasions, in 43 apes was stable over time at the level of the group, whether individual differences in these traits were reliable, and whether they were stable within the individual/unaffected by internal factors. As before, I appreciate the authors' choice of data analytic approaches, and I thought the writing, and explanations of their methods, were largely clear.

My main concerns were as follows. First, the paper was method-oriented rather than problem-oriented, that is, they did not focus on the main research questions in the Introduction. Second, I had not understood what they meant about "stability of group-level performance" and why my

30understanding of what they meant was important. Third, I disagreed that their study addressed the question of whether the results of their study (and others) can be understood using associative learning. Finally, I had suggested several minor edits that they needed to address, including problems with figures.

With respect to my first concern, the Introduction has been improved, so maybe it's just me—and if it is, I apologize—but the it still seems more methods- than problem-oriented. I would just like the authors to present succinctly some statement about what possible explanation their study will exclude. It would be nice to see how they got to this question, too.

With respect to my second concern, the authors did a great job addressing it in the letter. I now see what they mean, although I am not certain that framing what they did as a replication (or something like it), as I seem to think they did (lines 162-163), is correct. Replication concerns repeating the study in new data (see <https://doi.org/10.1371/journal.pbio.3000691>).

I also still think there was too much on associative learning/simpler processes (my third concern) in the Discussion section (lines 438-450). It has been a while since I dug into the learning literature, but just because a response does not steadily increase or decreases over time does not mean that there is no simpler process. For instance, the authors cannot exclude the possibility that the decrease in gaze-following (lines 445-446) is an instance in which the animals learn to ignore a cue that is not ecologically valid *sensu* Tolman and Brunswick (<https://doi.org/10.1037/h0062156>). Related to this, latent learning (<https://doi.org/10.1037/h0062733>) may explain the findings reported on lines 446-447. As I stated before, the current study simply cannot exclude various alternatives and so the authors are best off not discussing the matter. Note that I realize my interpretation clashes with that of Reviewer 3, but I cannot see how these data speak to this question.

In addition to highlighting these matters, and some minor issues, I wanted to turn to how the authors addressed an important point raised by Reviewer 3 regarding other traits, for example those related to motivation. One place in which I presume the authors addressed this was in lines 301-303. Data from several studies on the association between personality traits and cognitive performance simply do not support the nomological net that the authors are trying to cast (off the top of my head, <https://doi.org/10.1098/rsos.170169>; <https://doi.org/10.12966/abc.02.04.2016>; <https://doi.org/10.7717/peerj.9707>; <https://doi.org/10.1037/a0031723>; <https://doi.org/10.1007/s10071-013-0603-5>).

My additional minor comments follow and I refer to the line numbers in the manuscript.

line 28: I would omit "uniquely" the phrase "uniquely human cognition" has become a cliché, in all honesty, and it seems redundant anyway.

line 29: I would delete "specific".

line 42: The use of "structured" here is awkward.

line 43: I think the word "abiding" can be omitted.

line 51: I'd move "directly" so that it follows "hominins" or delete it.

line 54: Insert "also" in this sentence and delete "now-".

line 63: Delete the hyphen between "non" and "human".

line 65: The word "several" here is redundant.

line 68: I think it's "researcher degrees of freedom".
line 71: Please omit the parentheses.
line 76: I'd say "the behavior" and omit "which is the basic material".
line 78: A reference is needed for the statement ending on this line.
lines 81-82: The text "asking these ... asking questions" is awkward, probably because "asking" is used twice here.
line 93: The phrase "in order to" can (and should) be written "to".
lines 94-95: The sentence spanning these lines repeats earlier text (lines 81-82), so consider revising some of the text on this page.
line 105: The phrase "undertook notable effort" is awkward.
lines 113-114: "Based on a..." should be "Using a..." You should also cite the Campbell and Fiske's 1959 paper from Psychological Bulletin here.
line 120: The phrase "general cognitive ability" shouldn't be in single quotes.
lines 122-123: I am not sure what this sentence is doing here. I would delete it.
line 124: Delete "seminal" please.
line 132: Delete "the nature of" please.
line 134: I think "directly" can be deleted.
lines 135-136: The text ending the sentence is awkward.
line 138: The word "across" can be deleted.
lines 142-145: This text is very awkward. It needs revising.
line 167: I think "percent correct" should be inserted after "mean".
line 168: "Structural Equation Modeling" should be in sentence case, as should "Latent State models", and similar.
line 179: Should it read "performance was"?
line 192: Instead of "licensed", use "supported", "were consistent with", or similar.
line 198: Insert "(total)" after "observed".
line 200: I would delete "(theoretically)".
line 207: I think another em dash is needed.
line 212: I think "two" can be deleted.
line 214: I would split the sentence at the end of this line.
line 219: A "full" what?
line 226: What "stands out"?
line 242: Fluctuations in what?
line 244: The phrase "tease them apart" is a cliché and has been used before here, I think, so change it, please.
line 251: The text on this line is awkward.
lines 346-347: This sentence is awkward.
line 391: Change "vs." to "and".
lines 470-471: The names of the species should not be capitalized. Also, should the species scientific names be noted?
line 475: I would delete the word "groups" within the parentheses and also the word "strictly".
lines 483-487: I think this text belongs under "Procedure" and the "Material" heading can be omitted.

Thank you for asking me to review this manuscript. I hope the authors find my comments helpful.

Alexander Weiss [I sign my reviews]

Reviewer #4:

Remarks to the Author:

I appreciate the revision. The authors have addressed most of my points from the previous round and I have no further comments.

Author Rebuttal, Second Revision:Reviewer #1

This is a revision of a manuscript. The authors tested whether several cognitive traits, each measured on multiple occasions, in 43 apes was stable over time at the level of the group, whether individual differences in these traits were reliable, and whether they were stable within the individual/unaffected by internal factors. As before, I appreciate the authors' choice of data analytic approaches, and I thought the writing, and explanations of their methods, were largely clear.

My main concerns were as follows. First, the paper was method-oriented rather than problem-oriented, that is, they did not focus on the main research questions in the Introduction. Second, I had not understood what they meant about "stability of group-level performance" and why my understanding of what they meant was important. Third, I disagreed that their study addressed the question of whether the results of their study (and others) can be understood using associative learning. Finally, I had suggested several minor edits that they needed to address, including problems with figures.

With respect to my first concern, the Introduction has been improved, so maybe it's just me—and if it is, I apologize—but the it still seems more methods- than problem-oriented. I would just like the authors to present succinctly some statement about what possible explanation their study will exclude. It would be nice to see how they got to this question, too.

Response: We are glad we could improve the introduction to make the focus of our study more clear. In the revised version, we now include a sentence that specifies an alternative view that can be ruled out by our data (page 6). We go to the question we address in this study by reflecting on the assumptions underlying much of primate cognition research. Relatedly, we were worried about to what extent the methods commonly used are suited to answer the questions researchers ask.

With respect to my second concern, the authors did a great job addressing it in the letter. I now see what they mean, although I am not certain that framing what they did as a replication (or something like it), as

I seem to think they did (lines 162-163), is correct. Replication concerns repeating the study in new data (see <https://doi.org/10.1371/journal.pbio.3000691>).

Response: We completely agree that replications involve new data. We apologise if our framing was unclear. We have removed the term “replicated” from the manuscript in the revised version.

*I also still think there was too much on associative learning/simpler processes (my third concern) in the Discussion section (lines 438-450). It has been a while since I dug into the learning literature, but just because a response does not steadily increase or decreases over time does not mean that there is no simpler process. For instance, the authors cannot exclude the possibility that the decrease in gaze-following (lines 445-446) is an instance in which the animals learn to ignore a cue that is not ecologically valid *sensu* Tolman and Brunswick (<https://doi.org/10.1037/h0062156>). Related to this, latent learning (<https://doi.org/10.1037/h0062733>) may explain the findings reported on lines 446-447. As I stated before, the current study simply cannot exclude various alternatives and so the authors are best off not discussing the matter. Note that I realize my interpretation clashes with that of Reviewer 3, but I cannot see how these data speak to this question.*

Response: Thank you for engaging in this discussion. We completely agree that our study was not designed to rule out simpler processes. The original publications, from which we adapted our tasks, unusually did that already. We also agree that our data is open to alternative interpretations regarding learning processes over time. We clearly state this at the end of the contested section and invite researchers to use our – publicly available – data to test such processes. Nevertheless, in agreement with Reviewer 3, we think it is worth making the point because the view that associative learning leads to a steady increase in performance over time is a, though overly simplistic, commonly held view in the field. Thus, we would prefer to keep the section in the paper.

In addition to highlighting these matters, and some minor issues, I wanted to turn to how the authors addressed an important point raised by Reviewer 3 regarding other traits, for example those related to motivation. One place in which I presume the authors addressed this was in lines 301-303. Data from several studies on the association between personality traits and cognitive performance simply do not support the nomological net that the authors are trying to cast (off the top of my head, <https://doi.org/10.1098/rsos.170169>; <https://doi.org/10.12966/abc.02.04.2016>; <https://doi.org/10.7717/peerj.9707>; <https://doi.org/10.1037/a0031723>; <https://doi.org/10.1007/s10071-013-0603-5>).

Response: Thank you for pointing this out. We can see how the wording in the previous version of the manuscript was misleading and we changed the revised version to be more descriptive. What we wanted to say is that stable differences in e.g. motivation to participate in studies would exert a stable influence on performance that would not vary from timepoint to timepoint. We also added references to some of the papers – thank you very much for providing the links.

My additional minor comments follow and I refer to the line numbers in the manuscript.

Response: Thank you very much for the detailed reading of our manuscript. We are very grateful for the feedback, which has significantly improved its quality.

line 28: I would omit “uniquely” the phrase “uniquely human cognition” has become a cliché, in all honesty, and it seems redundant anyway.

Response: The term “uniquely human” no longer appears in the abstract.

line 29: I would delete “specific”.

Response: The word has been removed in the new version of the abstract.

line 42: The use of “structured” here is awkward.

Response: We think it makes sense to use the term structured here because different cognitive abilities and the relations between them define Great Ape cognition.

line 43: I think the word “abiding” can be omitted.

Response: We changed the text accordingly.

line 51: I’d move “directly” so that it follows “hominins” or delete it.

Response: We changed the text accordingly.

line 54: Insert “also” in this sentence and delete “now-”.

Response: We changed the text accordingly.

line 63: Delete the hyphen between “non” and “human”.

Response: We changed the text accordingly.

line 65: The word “several” here is redundant.

Response: This section has been removed.

line 68: I think it’s “researcher degrees of freedom”.

Response: This section has been removed.

line 71: Please omit the parentheses.

Response: We changed the text accordingly.

line 76: I'd say "the behavior" and omit "which is the basic material".

Response: We changed the text accordingly.

line 78: A reference is needed for the statement ending on this line.

Response: We added a reference; thank you for pointing this out.

lines 81-82: The text "asking these ... asking questions" is awkward, probably because "asking" is used twice here.

Response: We changed the text.

line 93: The phrase "in order to" can (and should) be written "to".

Response: We changed the text accordingly.

lines 94-95: The sentence spanning these lines repeats earlier text (lines 81-82), so consider revising some of the text on this page.

Response: We think this repetition is warranted because we are making concrete here what we only hinted at above. We would thus prefer to keep it as is.

line 105: The phrase "undertook notable effort" is awkward.

Response: We removed the phrase.

lines 113-114: "Based on a..." should be "Using a..." You should also cite the Campbell and Fiske's 1959 paper from Psychological Bulletin here.

Response: We changed the text and added the citation.

line 120: The phrase "general cognitive ability" shouldn't be in single quotes.

Response: We changed the quotes.

lines 122-123: I am not sure what this sentence is doing here. I would delete it.

Response: The sentence points to previous work that sought to link individual differences in cognitive abilities to external variables. We thought it would be fair to cite this work because we are doing something similar in our analysis.

line 124: Delete "seminal" please.

Response: We deleted the word.

line 132: Delete “the nature of” please.

Response: We deleted the phrase.

line 134: I think “directly” can be deleted.

Response: We deleted the word.

lines 135-136: The text ending the sentence is awkward.

Response: We changed the ending of the sentence.

line 138: The word “across” can be deleted.

Response: We deleted the word.

lines 142-145: This text is very awkward. It needs revising.

Response: We edited the text to be more clear.

line 167: I think “percent correct” should be inserted after “mean”.

Response: We added this information.

line 168: “Structural Equation Modeling” should be in sentence case, as should “Latent State models”, and similar.

Response: We changed the spelling accordingly.

line 179: Should it read “performance was”?

Response: We changed the text accordingly.

line 192: Instead of “licensed”, use “supported”, “were consistent with”, or similar.

Response: We changed the text accordingly.

line 198: Insert “(total)” after “observed”.

Response: We inserted the word.

line 200: I would delete "(theoretically)".

Response: We removed the word

line 207: I think another em dash is needed.

Response: We added it.

line 212: I think "two" can be deleted.

Response: We removed the word.

line 214: I would split the sentence at the end of this line.

Response: We split the sentence.

line 219: A "full" what?

Response: The full test, sorry, we added the missing word.

line 226: What "stands out"?

Response: We changed the wording.

line 242: Fluctuations in what?

Response: In ranks – we added the missing words.

line 244: The phrase "tease them apart" is a cliché and has been used before here, I think, so change it, please.

Response: We removed the sentence in question.

line 251: The text on this line is awkward.

Response: We changed the wording.

lines 346-347: This sentence is awkward.

Response: We reformulated the sentence.

line 391: Change "vs." to "and".

Response: We removed the sentence to reach the word limit.

lines 470-471: The names of the species should not be capitalized. Also, should the species scientific names be noted?

Response: We changed the spelling and added the scientific names.

line 475: I would delete the word "groups" within the parentheses and also the word "strictly".

Response: We changed the text accordingly.

lines 483-487: I think this text belongs under "Procedure" and the "Material" heading can be omitted.

Response: We moved the section to Procedure and deleted Material.

Final Decision Letter:

Dear Professor Holtmann,

Please find below a copy of the decision letter for your manuscript "Great ape cognition is structured by stable cognitive abilities and predicted by developmental conditions" [NATECOLEVOL-220817321C], which has just been accepted for publication in Nature Ecology & Evolution.

As soon as your article is published, you can generate your shareable link by entering the DOI of your article here: <http://authors.springernature.com/share>. Corresponding authors will also receive an automated email with the shareable link.

Sincerely,

Editorial Assistant
Nature Ecology & Evolution
The Macmillan Building
4 Crinan Street
London, N1 9XW
UK

Subject: Decision on Nature Ecology & Evolution manuscript NATECOLEVOL-220817321C

28th March 2023

Dear Dr Bohn,

We are pleased to inform you that your Article entitled "Great ape cognition is structured by stable cognitive abilities and predicted by developmental conditions", has now been accepted for publication in Nature Ecology & Evolution.

41Over the next few weeks, your paper will be copyedited to ensure that it conforms to Nature Ecology and Evolution style. Once your paper is typeset, you will receive an email with a link to choose the appropriate publishing options for your paper and our Author Services team will be in touch regarding any additional information that may be required

You will not receive your proofs until the publishing agreement has been received through our system

Due to the importance of these deadlines, we ask you please us know now whether you will be difficult to contact over the next month. If this is the case, we ask you provide us with the contact information (email, phone and fax) of someone who will be able to check the proofs on your behalf, and who will be available to address any last-minute problems . Once your paper has been scheduled for online publication, the Nature press office will be in touch to confirm the details.

Acceptance of your manuscript is conditional on all authors' agreement with our publication policies (see www.nature.com/authors/policies/index.html). In particular your manuscript must not be published elsewhere and there must be no announcement of the work to any media outlet until the publication date (the day on which it is uploaded onto our web site).

Please note that *Nature Ecology & Evolution* is a Transformative Journal (TJ). Authors may publish their research with us through the traditional subscription access route or make their paper immediately open access through payment of an article-processing charge (APC). Authors will not be required to make a final decision about access to their article until it has been accepted. [Find out more about Transformative Journals](https://www.springernature.com/gp/open-research/transformative-journals)

Authors may need to take specific actions to achieve [compliance](https://www.springernature.com/gp/open-research/funding/policy-compliance-faqs) with funder and institutional open access mandates. If your research is supported by a funder that requires immediate open access (e.g. according to [Plan S principles](https://www.springernature.com/gp/open-research/plan-s-compliance)) then you should select the gold OA route, and we will direct you to the compliant route where possible. For authors selecting the subscription publication route, the journal's standard licensing terms will need to be accepted, including [self-archiving-and-license-to-publish](https://www.nature.com/nature-portfolio/editorial-policies/self-archiving-and-license-to-publish). Those licensing terms will supersede any other terms that the author or any third party may assert apply to any version of the manuscript.

We welcome the submission of potential cover material (including a short caption of around 40 words) related to your manuscript; suggestions should be sent to Nature Ecology & Evolution as electronic files (the image should be 300 dpi at 210 x 297 mm in either TIFF or JPEG format). Please note that such pictures should be selected more for their aesthetic appeal than for their scientific content, and that colour images work better than black and white or grayscale images. Please do not try to design a cover with the Nature Ecology & Evolution logo etc., and please do not submit composites of images related to your work. I am sure you will understand that we cannot make any promise as to whether any of your suggestions might be selected for the cover of the journal.

You can generate the link yourself when you receive your article DOI by entering it here: <http://authors.springernature.com/share>.

[REDACTED]

P.S. Click on the following link if you would like to recommend Nature Ecology & Evolution to your librarian <http://www.nature.com/subscriptions/recommend.html#forms>

** Visit the Springer Nature Editorial and Publishing website at http://editorial-jobs.springernature.com?utm_source=ejp_NEcoE_email&utm_medium=ejp_NEcoE_email&utm_campaign=ejp_NEcoE for more information about our career opportunities. If you have any questions please click [here](mailto:editorial.publishing.jobs@springernature.com).**